# AXBENCH: Steering LLMs? Even Simple Baselines Outperform Sparse Autoencoders

Zhengxuan Wu [* 1]  Aryaman Arora [* 1]  Atticus Geiger [2]  Zheng Wang [1]  Jing Huang [1]
Dan Jurafsky [1]  Christopher D. Manning [1]  Christopher Potts [1]

## Abstract

Fine-grained steering of language model outputs is essential for safety and reliability. Prompting and finetuning are widely used to achieve these goals, but interpretability researchers have proposed a variety of representation-based techniques as well, including sparse autoencoders (SAEs), linear artificial tomography, supervised steering vectors, linear probes, and representation finetuning. At present, there is no benchmark for making direct comparisons between these proposals. Therefore, we introduce AXBENCH, a large-scale benchmark for steering and concept detection, and report experiments on Gemma-2-2B and 9B. For steering, we find that prompting outperforms all existing methods, followed by finetuning. For concept detection, representation-based methods such as difference-in-means, perform the best. On both evaluations, SAEs are not competitive. We introduce a novel weakly-supervised representational method (Rank-1 Representation Finetuning; ReFT-r1), which is competitive on both tasks while providing the interpretability advantages that prompting lacks. Along with AXBENCH, we train and publicly release SAE-scale feature dictionaries for ReFT-r1 and DiffMean.

github.com/stanfordnlp/axbench

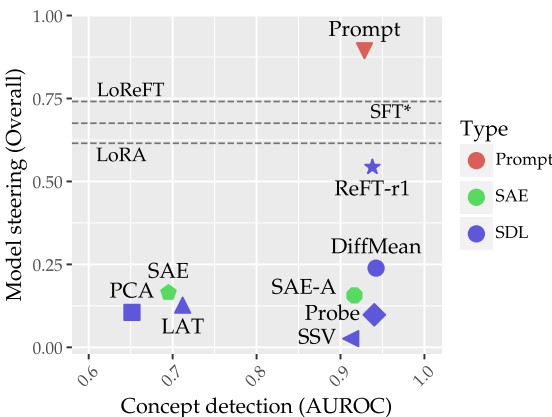

Figure 1: Average results across eight tasks on **C** concept detection (0–2) vs. **S** model steering (0–2) for all methods on AXBENCH. *Only evaluated on Gemma-2-2B.

## 1. Introduction

In order to be useful, language models (LMs) must follow user instructions and be aligned to human goals and values. While prompting and finetuning are now widely used to instill such behaviour in LMs, both methods have limitations: circumvention via jailbreaks and continued training, reliance on dataset quality, and uninterpretability (Anwar et al., 2024). Interpretability researchers have thus proposed a new class of representation-based interventions for **steering** LMs, which hope to address these issues. These methods include learning steering vectors from small labelled datasets, self-supervised sparse autoencoders (SAEs), among other techniques. Since steering may enable lightweight and interpretable control over model outputs, it has emerged as a potential alternative to finetuning and prompting (see §2).

Unfortunately, Pres et al. (2024); Braun et al. (2024) note that existing benchmarks for steering only evaluate a few methods at merely toy scales. To assess whether representation steering is a viable alternative to existing model control techniques, we need to evaluate it in a more realistic setting, e.g. over open-vocabulary concepts and on long-form generation, and compare it to prompting and finetuning baselines.

In this work, we introduce **AXBENCH**, a benchmark for evaluating LM control methods at scale using synthetic data.

*Equal contribution [1]Department of Computer Science, Stanford University [2]Pr(AI)[2]R Group. Correspondence to: Zhengxuan Wu <wuzhengx@cs.stanford.edu>, Aryaman Arora <aryaman@cs.stanford.edu>.

*Proceedings of the 42nd International Conference on Machine Learning*, Vancouver, Canada. PMLR 267, 2025. Copyright 2025 by the author(s).

AXBENCH takes in a list of natural language descriptions of concepts and samples relevant training and evaluation data from an LLM. We evaluate model-control methods, including prompting and finetuning baselines, along two utility **ax**es: **concept detection** C and **model steering** S. For the former, we use labelled synthetic data as ground truth; for the latter, we evaluate long-form generations using an LLM judge. The labelled training data enables comparison between supervised dictionary-learning methods (SDLs) and unsupervised methods like SAEs. The benchmark includes tasks generated from SAE concept lists for GemmaScope (Lieberum et al., 2024), covering two layers each from *instruction-tuned* Gemma-2-2B and Gemma-2-9B (Gemma Team et al., 2024). However, AXBENCH is by nature extensible to arbitrary concept descriptions: we intend to add new evaluation tasks as better feature-labelling techniques and new approaches to steering emerge.

We evaluate a variety of steering methods—including a novel weakly-supervised method we introduce, **ReFT-r1**—along with prompting, full finetuning, and two parameter-efficient finetuning methods (LoRA and LoReFT). On steering, only ReFT-r1 is competitive with finetuning and prompting baselines, while SAEs fall behind both ReFT-r1 and difference-in-means (Marks and Tegmark, 2024) on both axes. While representation steering methods largely lag behind incumbent model-control techniques, ReFT-r1 is evidence that steering can be pushed further with the availability of comprehensive evaluation benchmarks. Finally, along with AXBENCH, we train and publicly release SAE-scale feature dictionaries for ReFT-r1 and DiffMean.[1] We call this approach *supervised dictionary learning* (SDL; Figure 2).

## 2. Related work

**Representation-based control.** Interventional/causal interpretability has emerged as the dominant paradigm for understanding neural networks in the LLM era, enabling the reverse-engineering of circuits underlying specific behaviours (Giulianelli et al., 2018; Vig et al., 2020; Geiger et al., 2021; 2022; Meng et al., 2022; Chan et al., 2022; Wang et al., 2023; Goldowsky-Dill et al., 2023; Geiger et al., 2024; Guerner et al., 2024; Geiger et al., 2024). An important assumption in much of this work is the **linear representation hypothesis**, which claims that linear subspaces of representations in neural networks encode concepts (Mikolov et al., 2013b; Pennington et al., 2014; Bolukbasi et al., 2016; Elhage et al., 2022; Park et al., 2023; Nanda et al., 2023). Intervening on representations has thus emerged as an alternative to finetuning and prompting for LM control.

*Representation-based steering* by adding fixed vectors to

---

[1]We open-source all of our datasets and trained dictionaries at https://huggingface.co/pyvene.

activations, or clamping activations to a certain value along fixed directions, is one such intervention-based tool for model control (Zou et al., 2023; Li et al., 2024; Turner et al., 2024; Marks and Tegmark, 2024; Liu et al., 2024; van der Weij et al., 2024; Rimsky et al., 2024). Finetuning-based approaches such as ReFT (Wu et al., 2024a) enable optimisation of steering directions on a dataset. Steering vectors need not be computed from labelled data; SAEs enable scalable discovery of steering vectors from unlabelled data. In the same class of approaches, latent adversarial training (Casper et al., 2024) and circuit breakers (Zou et al., 2024) are representation-based control methods that increase the adversarial robustness of LLMs.

**Sparse autoencoders.** Sparse autoencoders (SAEs) aim to enable *self-supervised* and thus *scalable* decomposition of the representation space into meaningful concepts (Templeton et al., 2024; Chalnev et al., 2024; Makelov, 2024; O'Brien et al., 2024; Gao et al., 2024). SAEs are trained to reconstruct LLM hidden representations in a higher-dimensional latent space with a sparsity penalty, based on the assumption that concepts must be represented sparsely in order to prevent interference. The latents are then labelled with natural-language descriptions using automatic interpretability pipelines (e.g. Juang et al., 2024), which can then be used to identify useful latents to steer the LM.

Recent work reports mixed results when evaluating SAEs for steering; SAEs (but also several other steering methods) suffer from a tradeoff between model control and capabilities preservation (Mayne et al., 2024; Chalnev et al., 2024; Durmus et al., 2024; Bhalla et al., 2025). However, Karvonen et al. (2024) report Pareto-optimal performance when using SAEs to prevent models from producing regular expressions in code. Overall, evaluating SAEs remains an open problem because there is no ground-truth set of features to compare against.

## 3. AXBENCH

AXBENCH is a benchmark which takes in a list of natural language descriptions of concepts and synthetically generates the appropriate training and evaluation data for each concept using an LLM (Figure 2). The training and evaluation data consists of labelled pairs of instructions and responses, where the responses are either *positive* examples expressing the presence of the concept of interest, or *negative* examples that represent the unsteered behaviour of the model (see §3.1 for details).

We evaluate along two axes: **concept detection** C and **model steering** S. For the former, we measure classification performance on a held-out set of labelled data.[2] For the

---

[2]We focus on binarised concept detection, as a multi-class

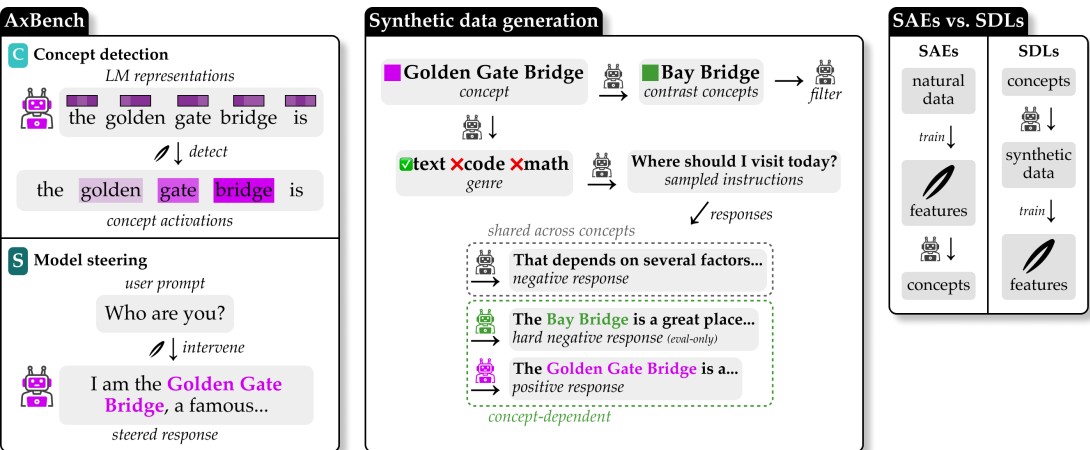

Figure 2: Overview of AXBENCH. **Left**—AXBENCH tasks: we evaluate *concept detection* by locating concept-specific activations and *model steering* by intervening on generations. **Center**—Synthetic data generation for the concept *Golden Gate Bridge*: we pick a contrast concept, filter by genre, sample instructions, and obtain concept-dependent responses. **Right**—Training pipelines: SAEs label pretrained features with LLM-derived concept tags, whereas SDLs prompt an LLM to synthesize training data and learn features directly from it.

latter, we use an LLM judge to rate steered outputs on three relevant axes (see §3.3).

In this work, we use natural language concept lists for GemmaScope SAEs as input, and generate training and evaluation data for the following representation sites: layers 10 and 20 of instruction-tuned Gemma-2-2B, and layers 20 and 31 of instruction-tuned Gemma-2-9B. We sample 500 concepts for each task to generate data; we term this dataset CONCEPT500. These eight tasks (4 sites × 2 axes) form the core training and evaluation testbeds for AXBENCH. Below, we describe the data generation process and evaluation setup for both axes.

### 3.1. Synthetic concept dataset generation

We construct a small training dataset $\mathcal{D}_{\text{train}} = \{(\mathbf{x}_{c,i}^{+}, y^{+})\}_{i=1}^{n/2} \cup \{(\mathbf{x}_{c,i}^{-}, y^{-})\}_{i=1}^{n/2}$ with $n$ examples and a concept detection evaluation dataset $\mathcal{D}_{\text{concept}}$ of the same structure and harder examples, where $y^{+}$ and $y^{-}$ are binary labels indicating whether the concept $\mathbf{c}$ is present. We set $n = 144$ for our main experiments.[3]

We query gpt-4o-mini-2024-07-18 to generate the data; the prompts used in this pipeline are presented in Appendix J.2. Generating the data requires the following steps (note that only the evaluation set includes hard negatives):

1. **Genre labelling & seed instructions**: We consider three genres: *text*, *code*, and *math*. We prompt the

---

classification task over $n$ classes can also be formulated into a binarised one over $n$ features.

[3]Using a small training dataset ensures our methods are practical and cost-effective alternatives to SAEs.

---

LLM to pick the genre $g_c$ for each concept.[4] We then randomly select seed instructions from our instruction pool which belong to genre $g_c$; see Appendix I for dataset details and for an explanation of how we sample instructions for different genres from existing public instruction datasets. We then prompt the LLM to generate responses to these instructions.[5]

2. **Positive examples**: For each randomly sampled instruction from the instruction pool, we prompt the LLM to generate a response that incorporates the concept $\mathbf{c}$. We use the generated concept-conditioned responses concatenated with their instructions (using the LM's chat template) as our positive set.

3. **Negative examples**: To evaluate the generalisation ability of each method, we independently sample seed instructions from all genres for negatives.[6] These instructions are shared across concepts in order to save generation costs (i.e., $(\mathbf{x}_{\mathbf{c}}^{-}, y^{-})_0^{n/2}$ is independent of the concept $\mathbf{c}$). We sample responses from the LM we plan to steer (not the LLM) without any additional instructions. We use the paired instructions and responses as our negative set.

4. **Hard negative examples** *(evaluation only)*: For each concept, we find contrasting concepts that are semantically related to our concept of interest but which should not activate the concept. We find these by (a) generat-

---

[4]Genre labelling increases input diversity. For example, inputs related to concepts such as *programming code contains syntactic errors* should contain code instead of descriptions of coding errors.

[5]Each example costs less than $0.00006.

[6]We sample instructions based on overall genre distribution: 70% from *text*, 15% from *code*, and 15% from *math*.

ing a list of phrases that are semantically relevant to our concept, (b) filtering for those which are polysemous, and (c) finding alternative senses of those words which our concept should not activate on. This results in a set of contrast concepts $c_{\text{contrast}}$, each of which is a specific sense of a polysemous word $w_{\text{contrast}}$. We then ask the LLM to generate responses incorporating $w_{\text{contrast}}$ into the sentence where $w_{\text{contrast}}$ should express the sense related to $c_{\text{contrast}}$. We use the contrastive responses paired with their instructions as our hard negative set.

The negative training set is not applicable to all methods (e.g. full finetuning only needs the positive training set for model steering).

### 3.2. 🇨 Concept detection

A popular LM interpretability method is to train *probes* (Conneau et al., 2018; Hewitt and Manning, 2019; Belinkov et al., 2017) that measure to what extent LM representations encode properties of interest, e.g. linguistic features. In recent years, the goal of concept detection has broadened to the open-vocabulary setting, with unsupervised methods becoming more common (Bills et al., 2023; Huben et al., 2024; Choi et al., 2024).

**Task description.** Formally, given a Transformer-based LM with a hidden dimension size of $d$, we define a concept classifier as a parameterized function $\Psi_{\text{Detect}}$ that maps a model representation $h \in \mathbb{R}^d$ into a *binary* label $\hat{y}$ indicating the relative presence of a concept:

$$\Psi_{\text{Detect}}(h) = \hat{y} \in \mathbb{R}^1 \tag{1}$$

where $\Psi$ is any function, e.g. a neural network.

**Evaluation dataset.** To evaluate a concept classifier, we measure how accurately it can predict ground-truth labels on the labelled evaluation set from $\mathcal{D}_{\text{concept}}$ (see §3.1).

**Evaluation metrics.** Since our labels are at the sequence-level, we need to aggregate token-label scores from $\Psi$ to evaluate it. Given a sequence of token representations $\mathbf{h}^l = [h_1^l, h_2^l, \ldots, h_n^l]$ with $n$ tokens at layer $l \in [1, m]$, we max-pool the detection scores to get a sequence-level prediction:

$$\hat{y}_{\text{Detect}} = \max\left(\Psi_{\text{Detect}}(\mathbf{h}^l)\right) \tag{2}$$

We then normalize $\hat{y}_{\text{Detect}}$ between $[0, 1]$ by min-max normalisation over the evaluation dataset for each concept. The predicted score represents how strongly a concept is present in a sequence, which we can compare to the true label.

### 3.3. 🇸 Model steering

Representation-based steering has emerged as a potential alternative to existing model-control methods (e.g. finetun-

ing and prompting) and a practical application of various interpretability methods (see §2). Unlike concept detection, model steering assesses *causal* efficacy in controlling model behaviour. Previous evaluation benchmarks for steering are not general-purpose; they either rely on a limited set of tasks (Zou et al., 2023; Makelov, 2024; Bhalla et al., 2025) or condition generation on a fixed prefix (Chalnev et al., 2024). To the best of our knowledge, we are the first to evaluate model steering methods in the open-vocabulary setting at scale.

**Task description.** Given a prompt $\mathbf{x}$, the model's original generation can be written as $\hat{\mathbf{y}} = \text{LM}(\mathbf{x})$. We produce the model's counterfactual generation conditioned on the concept-based intervention $\Phi_{\text{Steer}}(\mathbf{h})$:

$$\hat{\mathbf{y}}_{\text{Steer}} = \text{LM}\left(\mathbf{x}, \mathbf{h} \leftarrow \Phi_{\text{Steer}}(\mathbf{h})\right) \tag{3}$$

where $\mathbf{h} \leftarrow \Phi_{\text{Steer}}(\mathbf{h})$ is an in-place representation modification. We use the open-source intervention library pyvene to perform such interventions on PyTorch implementations of models (Wu et al., 2024b).

**Evaluation dataset.** We evaluate these steering methods in the instruction-following setting, where we sample instructions from Alpaca-Eval (Li et al., 2023) and prompt the LM to generate a response while intervening on its forward pass in-place using one of the steering methods.

**Evaluation metrics.** For the intervened model generation, we evaluate $\hat{\mathbf{y}}_{\text{Steer}}$ based on the *harmonic mean* of the following scores, each of which the LLM rates using a discrete score of 0, 1, or 2:

1. **Concept score** represents how well the concept is incorporated into the response.
2. **Instruct score** represents how well the response is related to the instruction.
3. **Fluency score** represents how fluent the response is.

Since we compute the harmonic mean, the overall score also ranges from 0 to 2, but heavily penalises poor performance on any of these three subscores. For each concept, we randomly sample 10 instructions from Alpaca-Eval and sample continuations for each steering factor (see discussion on steering factor in §5.2). To ensure a fair comparison, we partition our instructions into two equally sized sets, selecting the best factor from one set and evaluating it on the hold-out set. Our judge prompts with further discussion can be found in Appendix J.3. We also validate our LLM-generated overall score with human evaluation in Appendix M.

## 4. Methods

In this section, we describe the interpretability methods we evaluate along with our baseline prompting and finetun-

ing methods. For each method, we label which axes it is evaluated on using Ⓒ and Ⓢ. All of our interpretability methods except SAEs are SDLs that learn rank-1 subspaces for targeted concepts.

**Notation.** Given a LM, the hidden representations of dimensionality $d$ for a token sequence of length $n$ in layer $l$ of the LM are represented as $\mathbf{h}^l = [h_1^l, h_2^l, \ldots, h_n^l] \in \mathbb{R}^{n \times d}$. The set of representations concatenated from all of the training set inputs is denoted as $\mathbf{H} \in \mathbb{R}^{s \times d}$, where $s = \sum_{\mathbf{h}} |\mathbf{h}|$. We denote $\mathbf{H}^+$ as the subset of $\mathbf{H}$ including only positive training inputs and $\mathbf{H}^-$ for the negative inputs (see §3.1 for training dataset details). Finally, per-method projection vectors $\mathbf{w}$ and representations $h_i$ are the same shape: $\mathbb{R}^{d \times 1}$.

Ⓒ Ⓢ **Difference-in-means (DiffMean).** DiffMean uses the difference between averaged representations from two classes of inputs as a steering vector (Marks and Tegmark, 2024). The projection vector $\mathbf{w}_{\text{DiffMean}}$ is defined as:

$$\mathbf{w}_{\text{DiffMean}} = \underbrace{\frac{1}{|\mathbf{H}^+|} \sum_{h_i^+ \in \mathbf{H}^+} h_i^+}_{\text{mean of positives}} - \underbrace{\frac{1}{|\mathbf{H}^-|} \sum_{h_i^- \in \mathbf{H}^-} h_i^-}_{\text{mean of negatives}} \quad (4)$$

We compute detection scores with the dot product, i.e. $\Psi_{\text{Detect}}^{\text{DiffMean}}(h_i) = h_i \cdot \mathbf{w}_{\text{DiffMean}}$.[7] Our steering operation is simple activation addition: $\Phi_{\text{Steer}}^{\text{DiffMean}}(h_i) = h_i + \alpha \mathbf{w}_{\text{DiffMean}}$ where $\alpha$ is the steering magnitude, which depends on the steering factor and is optimized as a hyperparameter, as described in §5.2.

Ⓒ Ⓢ **Principle component analysis (PCA).** For PCA, we use the first principal component of the positive set of hidden representations as the projection vector.[8] We first subtract the mean $\overline{\mathbf{H}^+}$ from each $h^+$, gathering the centered vectors into a matrix $\mathcal{H} \in \mathbb{R}^{|\mathbf{H}^+| \times d}$. We then find the top principal component $\mathbf{w}_{\text{PCA}} \in \mathbb{R}^{d \times 1}$ of $\mathcal{H}$, i.e. the unit vector that captures the largest variance along its direction, using sklearn.decomposition.PCA (Pedregosa et al., 2011). We follow the same detection and steering setup as DiffMean.

Ⓒ Ⓢ **Linear artificial tomography (LAT).** LAT searches for a single latent direction that can separate positive examples by learning from their pairwise activation differences (Zou et al., 2023). Concretely, we create pairwise activation differences $\delta$ by randomly partitioning $\mathbf{H}$ into pairs $(h_i, h_j)$ (with $i \neq j$) and computing $\delta = \frac{h_i - h_j}{\|h_i - h_j\|}$,

where the denominator ensures each difference is unit-normalized. We gather all these pairwise differences into a matrix $\Delta \in \mathbb{R}^{\frac{|\mathbf{H}|}{2} \times d}$. We then perform PCA (using sklearn) on $\Delta$; then $\mathbf{w}_{\text{LAT}} \in \mathbb{R}^{d \times 1}$ is the top principal component of $\Delta$. We follow the same detection and steering setup as DiffMean.

Ⓒ Ⓢ **Linear probe (Probe).** The linear probe learns to classify tokens as concept-relevant by projecting representations $h_i$ onto a learned direction $\mathbf{w}_{\text{Probe}} \in \mathbb{R}^{d \times 1}$ just as in DiffMean. To convert this into a probability, we apply the sigmoid activation, and then minimise binary cross-entropy loss with the true labels:

$$\min_{\mathbf{w}_{\text{Probe}}} \left\{ \frac{1}{|\mathbf{h}|} \sum_{h_i \in \mathbf{h}} \left( \mathcal{L}_{\text{BCE}}(y, \text{Sigmoid}(h_i \cdot \mathbf{w}_{\text{Probe}})) \right) \right\} \quad (5)$$

where $y$ is the token-level class label indicating whether this token belongs to a positive or negative example. The detection and steering setup is then identical to DiffMean.

Ⓒ Ⓢ **Supervised steering vector (SSV).** The supervised steering vector method directly learns an intervention that maximises the language-modelling probability of the positive responses. For a sequence of token representations $\mathbf{h}$, we apply an intervention to each token representation:

$$\Phi^{\text{SSV}}(h_i) = h_i + \mathbf{w}_{\text{SSV}} \quad (6)$$

where $\mathbf{w}_{\text{SSV}} \in \mathbb{R}^{d \times 1}$ is a learned vector. As described in §3.3, we backpropagate gradients by training with the language modeling loss, similar to supervised fine-tuning (SFT):

$$\min_{\mathbf{w}_{\text{SSV}}} \left\{ \sum_{t=1}^n \log P_{\text{LM}} \left( y_t \mid y_{<t}, \mathbf{x}; \mathbf{h} \leftarrow \Phi^{\text{SSV}}(\mathbf{h}) \right) \right\} \quad (7)$$

where $y_i$ is the $i$-th output token, $y_{<i}$ are the preceding tokens, and $\mathbf{x}$ is the prompt. For evaluating concept detection and model steering SSV follows the same setup as DiffMean. We apply ReLU to get the detection scores.

Ⓒ Ⓢ **Rank-1 representation finetuning (ReFT-r1).** We introduce a novel method based on ReFT (Wu et al., 2024a) which jointly learns concept detection and steering on supervised data by combining the training objectives of linear probing and supervised steering.

We compute latents for concept detection as:

$$\Psi_{\text{Detect}}^{\text{ReFT-r1}}(h_i) = \text{ReLU}(h_i \cdot \mathbf{w}_{\text{ReFT-r1}}) \quad (8)$$

During training we perform a representation-level intervention on each $h_i$ based on the latents of the sequence $\mathbf{h}$:

$$\Phi^{\text{ReFT-r1}}(h_i) = h_i + \left( \frac{1}{k} \left\| \text{TopK}(\Psi_{\text{Detect}}^{\text{ReFT-r1}}(\mathbf{h})) \right\|_1 \right) \mathbf{w}_{\text{ReFT-r1}} \quad (9)$$

---

[7]Following Gao et al. (2024), we normalize $\mathbf{w}_{\text{DiffMean}}$ to have unit norm. We apply the same normalization to the learned weights of PCA, LAT, Probe, and ReFT-r1.

[8]We found no significant difference between using only the positive set vs. the entire set of hidden representations for both PCA and LAT; see Appendix F for ablations.

where $\mathbf{w}_{\text{ReFT-r1}} \in \mathbb{R}^{d \times 1}$ is a learned vector. Finally, the training objective combines language modelling loss subject to this intervention, along with L1 regularisation on the non-top-$k$ latents:

$$\min_{\mathbf{w}_{\text{ReFT-r1}}} \left\{ -\sum_{t=1}^{n} \log P_{\text{LM}}^{\Phi^{\text{ReFT-r1}}}(y_t \mid y_{<t}, \mathbf{x}) + \lambda \sum_{a_i \notin \text{TopK}(\Psi(\mathbf{h}))} \|a_i\|_1 \right\} \tag{10}$$

Detection and steering is identical to DiffMean.

**C S Sparse autoencoders (SAE).** Sparse autoencoders are a self-supervised dictionary learning method (see §2). We use pretrained SAEs from GemmaScope, which are the best available SAEs for Gemma-family LLMs (Lieberum et al., 2024).[9] The SAEs we used are trained to learn two dictionary matrices, $\{\mathbf{W}_{\text{enc}}, \mathbf{W}_{\text{dec}}\} \in \mathbb{R}^{d \times z}$ where $z$ is the number of latents. For our evaluating concept $c$, we use $\{\mathbf{w}_{\text{enc}}, \mathbf{w}_{\text{dec}}\} \in \mathbb{R}^{d \times 1}$ as the detection and steering representations, respectively:

$$\Psi_{\text{Detect}}^{\text{SAE}}(h_i) = \sigma\left(h_i \cdot \mathbf{w}_{\text{enc}} + b_{\text{enc}}\right)$$

where $\sigma$ is an activation function (in our case, JumpReLU) and $b_{\text{enc}}$ is a learned bias.[10] For steering, we use activation addition as DiffMean. Note that Templeton et al. (2024) use activation clamping; we report ablations in Appendix F.

**C S SAEs with AUROC selection (SAE-A).** Given that other methods have access to a training dataset, to enable fair comparison we attempt to use our training dataset for SAE feature selection. For each feature, we compute its max-pooled activations per Equation (2) over each training example, compute AUROC over the dataset given true labels, and select the highest-scoring feature by this metric.

**C Bag-of-Words (BoW).** For the BoW baseline, we first construct a featurizer that tokenizes text by whitespace and counts word frequencies. The vocabulary for this featurizer is derived from the training dataset. We then train a logistic regression classifier to predict class probabilities, framing the task as binary classification. To mitigate overfitting, we incorporate a regularization term. This BoW approach leverages statistical biases inherent in LLM-generated data.

**C Gradient-based baselines.** We test two gradient-based attribution methods, which are applicable only to concept detection: Input × gradients ($\mathbf{I} \times \mathbf{G}$) and Integrated gradients

(**IG**; Sundararajan et al., 2017). For both, we train a classification head on the hidden representations of some layer and apply the methods to produce token-level attribution scores $\Psi_{\text{Detect}}(h_i)$. Implementation details are in Appendix H.

**C S Prompting baseline.** For concept detection, we use the same LLM judge as described in §3.3 to rate the presence of a concept on a scale of 0 to 2. For model steering, we use an LLM to *engineer* a prompt given a concept, which we use to steer our local model by prepending it to the actual instruction. We provide prompt templates and examples in Appendix J and Appendix O. [11]

**S Finetuning baselines.** We test full-parameter supervised finetuning (**SFT**) and two parameter-efficient finetuning methods: Low-rank adaptation (**LoRA**; Hu et al., 2022) and low-rank representation finetuning (**LoReFT**; Wu et al., 2024a). In all cases, we finetune to minimise the language-modelling loss on the responses in the positive split of the dataset; the negative training split is discarded. We then use the finetuned models as baselines for steering.

For all of our SDLs except SSV, we constrain any learned subspace to have a unit norm, following the same setup as SAEs. With a unit-norm constraint, we find that SSV is hard to use for steering models. For prompting and finetuning baselines, we randomly score one generation on the testing instruction set (since the factor is not a parameter for those methods), resulting in the same number of observations for those methods.

### 4.1. Evaluation

**Datasets.** We synthetically generate training and validation datasets (see §3.1) for 500 concepts, which we release as CONCEPT500. The concepts are sampled from the Neuronpedia SAE concept list for GemmaScope as described in Appendix B. For each concept, we include 144 examples for training and ≈72 samples for evaluating concept detection.[12] In this paper, we train and evaluate all methods, and report results on CONCEPT500. For SFT, we only train and evaluate on the first 20 concepts due to limited resources.

For evaluating steering, we use the instructions from the Alpaca-Eval dataset (Li et al., 2023). For each concept, we sample 10 instructions. We generate up to 128 tokens for each instruction over 14 steering factors. We split the instructions into two equal sets – one for selecting the best factor and the other for evaluation.

We additionally release training and evaluation datasets

---

[9]GemmaScope releases a set of SAEs for Gemma-2-27B, but the concept list is not publicly released, which makes the SAEs for Gemma-2-9B the largest ones available for evaluations.

[10]Note that this parameterisation cannot apply to TopK (Gao et al., 2024) and BatchTopK SAEs, which require loading in the entire encoder matrix to compute latents.

[11]We also experiment with a template-based zero-shot steering prompt and observe only slightly lower performance ($\pm 0.02$) on a small set of concepts.

[12]This varies based on valid hard negatives.

| Method | Gemma-2-2B | | Gemma-2-9B | | Avg. |
| | L10 | L20 | L20 | L31 | |
|---|---|---|---|---|---|
| DiffMean | 0.948 | 0.946 | 0.955 | 0.921 | **0.942** |
| Probe | 0.940 | 0.946 | 0.933 | **0.942** | 0.940 |
| ReFT-r1 | **0.952** | **0.965** | **0.966** | 0.869 | 0.938 |
| Prompt | 0.910 | 0.921 | 0.940 | 0.943 | 0.929 |
| SAE-A | 0.924 | 0.911 | 0.924 | 0.907 | 0.917 |
| BoW | 0.909 | 0.931 | 0.904 | 0.912 | 0.914 |
| SSV | 0.934 | 0.950 | 0.910 | 0.854 | 0.912 |
| LAT | 0.742 | 0.809 | 0.572 | 0.725 | 0.712 |
| SAE | 0.735 | 0.755 | 0.631 | 0.659 | 0.695 |
| PCA | 0.714 | 0.712 | 0.559 | 0.622 | 0.652 |
| IG | 0.440 | 0.375 | 0.508 | 0.383 | 0.426 |
| IxG | 0.243 | 0.217 | 0.193 | 0.330 | 0.246 |

Table 1: 🄲 Mean AUROC for each method on concept detection. **Bold** indicates highest AUROC in that column; underline indicates no significant difference vs. the best performer. Gray indicates non-representation steering methods.

for all 16K concepts in GemmaScope as the CONCEPT16K dataset suite. We train and release SAE-scale dictionaries on this dataset only for the best-performing methods found on CONCEPT500. See Appendix L for dataset statistics and Appendix E for further experiments on CONCEPT16K.

**Models.** Our evaluations rely on access to and control over the LLM's representations. To reduce training cost, we prefer to use models for which pretrained SAEs are available. We thus evaluate our methods on two open models, Gemma-2-2B-it and Gemma-2-9B-it (henceforth referred to without the -it suffix), from the Gemma-family, with corresponding SAEs released as GemmaScope. We evaluate our methods with model representations from the residual streams of layers 10 and 20 for Gemma-2-2B and layers 20 and 31 for Gemma-2-9B. We use SAEs from GemmaScope that are trained for these layers.[13] To ensure a fair comparison, we perform separate hyperparameter-tuning for each method. Details can be found in Appendix K.

## 5. Results

### 5.1. 🄲 Concept detection

For concept detection, CONCEPT500 consists of passages of text with ground-truth labels for each concept. Each method provides us with token-level concept scores obtained from the representation of that token at a particular layer. To compute a passage-level score, we take the mean of the token-level concept scores. See Appendix N for a visualization of token-level concept scores.

---

[13]For Gemma-2-2B, we follow the common practice to use SAEs for the base LM, as SAEs are not available for the instruction-tuned model at the time of publication (Lieberum et al., 2024).

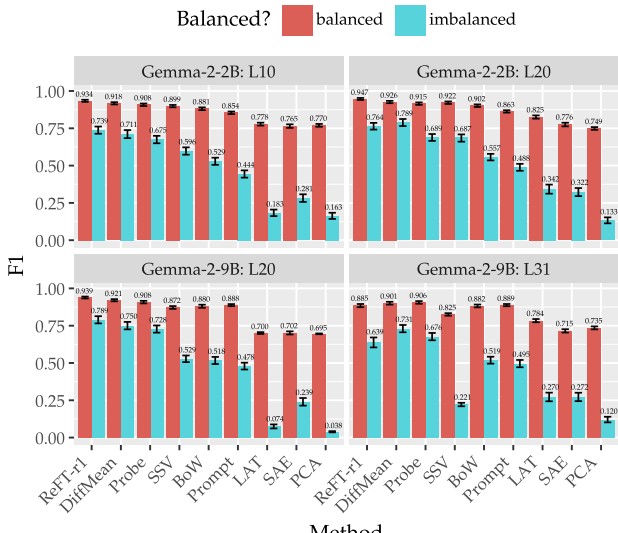

Figure 3: 🄲 Mean F1 scores vs. dataset balance.

**AUROC.** In Table 1, we report the average area under the ROC curve (AUROC) for each method over all concepts. Overall, we find that DiffMean, Probe, and ReFT-r1 are the best performers with no statistically significant difference ($p < 0.05$) between any of them under a paired $t$-test. Prompt, SAE-A, and SSV are not far behind and significantly outperform the remaining methods. LAT also performs better than random. Vanilla SAEs are thus significantly outperformed by five supervised methods, all of which are much cheaper to train using a limited amount of synthetic data. The remaining methods (PCA, IG, and IxG) perform poorly; PCA's better-than-random performance is nevertheless impressive given its unsupervised nature. Additional results are given in Appendix C.

**F1 score under class imbalance.** In real-world text, positive instances of concepts are much rarer than negative instances. We thus report F1 on both the balanced setting (50% positive instances) and an imbalanced setting with 3600 additional negative examples ($\approx$1% positive). We choose classification threshold by maximising F1, binarise the resulting predictions, and report statistics on this discrete classification. Figure 3 shows that the relative ordering of methods does not change substantially between the two settings; despite their sparsity, SAEs perform poorly, but LAT and PCA also degrade substantially.

### 5.2. 🅂 Model steering

For model steering, we take concept labels from CONCEPT500 and apply the (pre)trained steering methods to the base model and sample generations. We score the generations using an LM judge as described in §3.3. We addi-

| Method | Gemma-2-2B | | Gemma-2-9B | | Avg. |
|---|---|---|---|---|---|
| | L10 | L20 | L20 | L31 | |
| Prompt | 0.698 | **0.731** | **1.075** | **1.072** | **0.894** |
| LoReFT | **0.701** | 0.722 | 0.777 | 0.764 | 0.741 |
| SFT | 0.637 | 0.714 | — | — | *0.676* |
| LoRA | 0.637 | 0.641 | 0.602 | 0.580 | 0.615 |
| ReFT-r1 | 0.633 | 0.509 | 0.630 | 0.401 | 0.543 |
| DiffMean | 0.297 | 0.178 | 0.322 | 0.158 | 0.239 |
| SAE | 0.177 | 0.151 | 0.191 | 0.140 | 0.165 |
| SAE-A | 0.166 | 0.132 | 0.186 | 0.143 | 0.157 |
| LAT | 0.117 | 0.130 | 0.127 | 0.134 | 0.127 |
| PCA | 0.107 | 0.083 | 0.128 | 0.104 | 0.105 |
| Probe | 0.095 | 0.091 | 0.108 | 0.099 | 0.098 |
| SSV | 0.072 | 0.001 | 0.024 | 0.008 | 0.026 |

Table 2: **S** Mean overall steering scores for each method, after steering factor selection. Gray indicates non-representation steering methods.

| Method | Gemma-2-2B | | Gemma-2-9B | | Avg. |
|---|---|---|---|---|---|
| | L10 | L20 | L20 | L31 | |
| Prompt | **90.0%** | **91.5%** | **97.6%** | **99.1%** | **94.5%** |
| LoReFT | 88.9% | 88.2% | 88.6% | 90.3% | 89.0% |
| SFT | **90.0%** | 87.5% | — | — | *88.8%* |
| LoRA | 85.0% | 83.4% | 79.9% | 81.5% | 82.5% |
| ReFT-r1 | 85.2% | 82.3% | 83.6% | 76.0% | 81.8% |
| DiffMean | 63.2% | 55.2% | 64.3% | 52.2% | 58.7% |
| SAE | 50.0% | 50.0% | 50.0% | 50.0% | 50.0% |
| SAE-A | 49.3% | 46.6% | 48.5% | 50.7% | 48.8% |
| LAT | 43.5% | 48.2% | 42.7% | 48.6% | 45.8% |
| PCA | 42.1% | 42.9% | 42.2% | 45.4% | 43.1% |
| Probe | 40.4% | 44.0% | 41.9% | 45.6% | 43.0% |
| SSV | 38.8% | 32.0% | 32.5% | 34.0% | 34.3% |

Table 3: **S** Winrate against SAEs for each method, after steering factor selection.

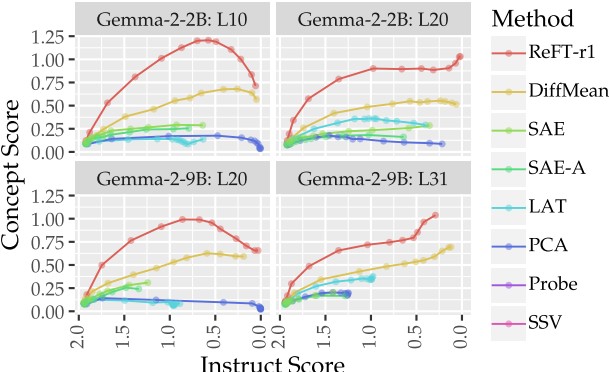

Figure 4: **S** Mean concept score vs. instruct score as the steering factor for each method is varied.

tionally benchmark prompting, full-finetuning (SFT), and two parameter-efficient finetuning methods (LoReFT and LoRA) as non-steering baselines.

For steering methods, we note that steering factor is an important hyperparameter. We select the optimal steering factor for each method independently for every concept based on which factor achieves the highest *overall* steering score, as given by the LLM judge. Our actual steering magnitude (i.e., $\alpha$, as described in §4) is the product of the steering factor and the maximal activations aggregated over the evaluation dataset for concept detection.[14]

**Overall scores.** We report the mean overall score for each method (i.e. the harmonic mean of three subscores: fluency, instruction-following, and concept presence) in Table 2. Prompting, along with slightly worse finetuning baselines, outperforms all steering methods on average, except

---

[14]For SAEs, we query Neuronpedia to obtain the maximal activation per concept.

for ReFT-r1. ReFT-r1 is competitive with prompting in Gemma-2-2B but significantly behind on Gemma-2-9B; prompting scores improve by a large margin on the larger model. Additionally, DiffMean significantly outperforms SAEs, particularly in earlier layers.

The remaining supervised steering methods fail to beat SAEs, and no steering methods besides ReFT-r1 approach prompting or finetuning performance. Importantly, we note that SAE-A slightly underperforms the unsupervised SAE; better classification does not directly lead to better steering.

**Winrate.** We compute winrates against SAEs by comparing overall scores on each concept under each setting. We treat ties as 0.5 wins and 0.5 losses. We report the results in Table 3. Again, ReFT-r1 (88.0%) and DiffMean (61.6%) achieve winrates of greater than 50% against SAEs, and relative rankings are similar to those for overall score. We note that DiffMean and ReFT-r1 show higher winrates on earlier layers in both models.

**Steering factor.** We compare the effect of changing the steering factor on instruct vs. concept scores in Figure 4. We notice that increasing the factor monotonically reduces instruct score in all methods, i.e. larger steering vectors harm capabilities; this agrees with prior findings (Durmus et al., 2024; Chalnev et al., 2024). However, the effect varies by layer for concept score: concept score increases then decreases in earlier layers, while it roughly monotonically increases with steering factor in later layers. In all cases, ReFT-r1 traces a Pareto-optimal path, achieving the highest concept score for any chosen instruct score.

## 6. Discussion

**Simple yet powerful baselines.** While representation-level interventions have been shown to be useful in both

enhancing model capabilities and for safety (see §2), they fail to outperform standard prompting and finetuning baselines on AXBENCH. This is sobering evidence of the current limitations of steering techniques. However, our results suggest that joint learning of concept detection and steering (as in ReFT-r1) may be the key to advancement.

**SDL vs. SAEs.** We have shown that SDL methods can achieve similar scalability and better performance at a lower cost compared to SAEs. Unlike SAEs, SDL methods require concepts to be known *a priori*; however, SDLs can be easily augmented with new features without retraining. We also note that SDLs depend on high-quality data generators, whereas SAEs rely on high-quality concept discriminators. These methods are not mutually exclusive and can complement each other.

**SAE concept label quality.** The concept lists used in this paper were adapted from Neuronpedia's auto-interpretability pipeline, which is often skewed towards token-level concepts and misses high-level abstractions. While we tried to do post-hoc SAE feature selection to mitigate this, the poor performance of SAEs is at least partially a reflection of the limitations of auto-interpretability. It would be interesting to explore whether the SAE performance on AXBENCH improves as better feature labelling methods are used and labels become less shallow (e.g. Choi et al., 2024). We conduct a preliminary study assessing the steering performance of SAEs and SDLs using a limited set of higher-quality concept labels (see Appendix E.4). Our results show that, although higher-quality labels improve SAE performance, they do not narrow its gap to SDLs such as ReFT-r1. Future work might explore rule-based concepts, as in generic instruction-following benchmarks such as IFEval (Zhou et al., 2023).

## 7. Conclusion

We introduced AXBENCH, a new benchmark for evaluating LM control methods at scale using synthetic data. To answer the question in the title of this work: our evaluation shows that even at SAE scale, representation steering is still *far behind* simple prompting and finetuning baselines. Simultaneously, we showed that a novel steering method, ReFT-r1, is capable of *closing the gap* to some extent; representation-based steering has not yet exhausted its potential. No matter the outcome, we believe that comprehensive evaluation benchmarks like AXBENCH are necessary for continued progress on this problem.

## Impact Statement

In this paper, we explore representation-based methods for steering language models and introduce AXBENCH, a large-scale benchmark for evaluating these techniques. We believe that the immediate ethical and societal implications of our research are minimal. However, we recognize that enhanced control over language model outputs could potentially be misused to reinforce biases or manipulate information. To address these concerns, we advocate for the responsible application of steering methods and ensure transparency by publicly releasing our datasets and feature dictionaries. We encourage ongoing collaboration and dialogue within the research community to monitor and mitigate any unintended consequences of these technologies.

## Acknowledgements

We thank Róbert Csordás, Qinan Yu, and Jiuding Sun for constant and extremely helpful feedback during our weekly interp meetings; Jake Mendel for enlightening discussion about the direction and framing of the work; Neel Nanda for helpful suggestions on SAE feature selection; We thank all of our human annotators listed in Appendix M; and Chenglei Si, Ken Ziyu Liu, Oam Patel, Luke Bailey, Harshit Joshi, Yanzhe 'Sanju' Zhang, Nikil Roashan Selvam, Julie Kallini, Omar Shaikh, Thomas Chen, Tristan Thrush, and Yangjun Ruan for various helpful discussions. We thank Joseph Tey and Nick Jiang for pointing out equation typos in an earlier draft.

This research is supported in part by grants from Open Philanthropy.

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

# Appendix

## A. Historical notes on steering

*Inspired by Jurafsky and Martin (2025) and noting the sociological observations about (mechanistic) interpretability as a field in Saphra and Wiegreffe (2024), we offer some historical notes on the development of steering as a field in an effort to document and properly cite where these ideas came from.*

*Steering* refers to applying interventions (usually adding a fixed vector) to the activation space of a neural model in order to control its generations. Early precursors to steering noted that linear subspaces of the representation space of pretrained word vectors seemed to encode meaningful concepts (Mikolov et al., 2013a; Pennington et al., 2014; Bolukbasi et al., 2016).

Larsen et al. (2016) first used the *difference-in-means* technique to extract visual *attribute vectors* from GAN discriminators in order to steer generator outputs; this technique was widely adopted in computer vision (White, 2016; Upchurch et al., 2017; Goh, 2017; Wang et al., 2019).

In NLP, initial work by Subramani et al. (2022) proposed *steering vectors*, learned to maximise the probability of some output, as an alternative to expensive fine-tuning and unreliable prompt optimisation for the task of controllable text generation. Soon after, steering was also use to localise behaviours in a maze-searching RL agent (Turner et al., 2023a;b; Mini et al., 2023). Variations on this approach (sometimes using difference-in-means or other closed-form expressions to compute the vector) were adopted by researchers in *mechanistic interpretability* from late 2023 for AI safety (Zou et al., 2023; Li et al., 2024; Turner et al., 2024; Marks and Tegmark, 2024; Rimsky et al., 2024) and later as a general-purpose but localised and parameter-efficient alternative to finetuning (Wu et al., 2024a; Liu et al., 2024; van der Weij et al., 2024).

*Sparse autoencoders* (SAEs), a scalable technique for self-supervised rank-one linear feature discovery via dictionary learning, are also increasingly used to find or learn steering vectors (Templeton et al., 2024; Chalnev et al., 2024; Makelov, 2024; O'Brien et al., 2024).

## B. SAE concept list

We use SAE concept lists to enable a fair comparison with SAEs, which were annotated mostly by `gpt-3.5-turbo` or `gpt-4o-mini`. These concept lists are released by Neuronpedia and were scraped by the authors of this paper in November 2024. We utilize the concept lists from four SAEs from GemmaScope: `10-gemmascope-res-16k` for the Gemma-2-2B base model and `20-gemmascope-res-131k` for the Gemma-2-9B instruction-tuned model, where we scraped a maximum of 16K concepts.

# C. Detailed analysis

## C.1. C Concept detection

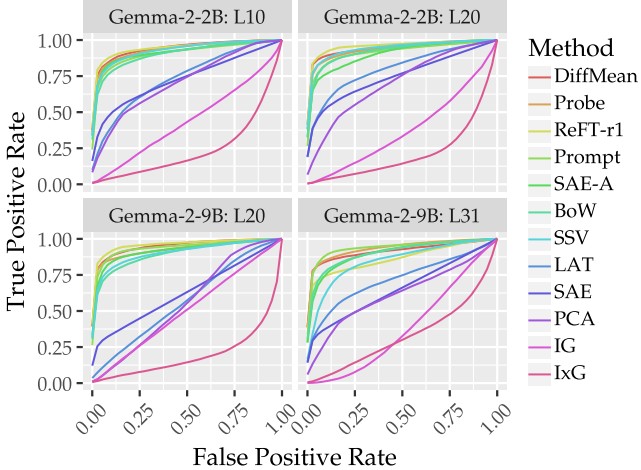

Figure 5: C Mean ROC curves over all concepts.

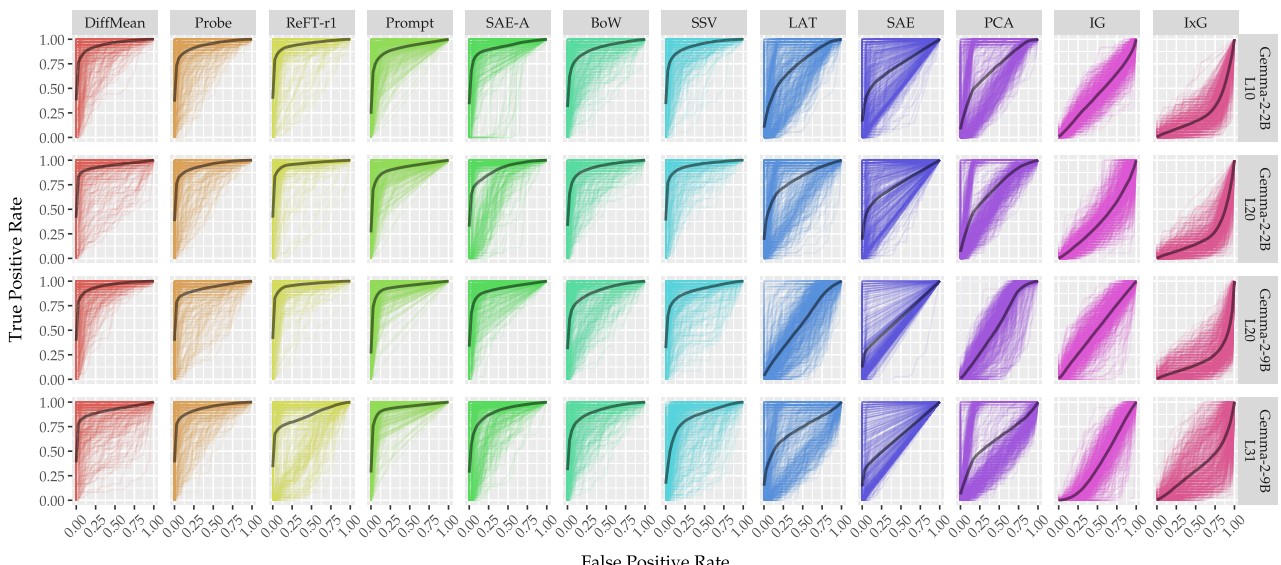

Figure 6: All ROC curves.

## C.2. Ⓢ Model steering

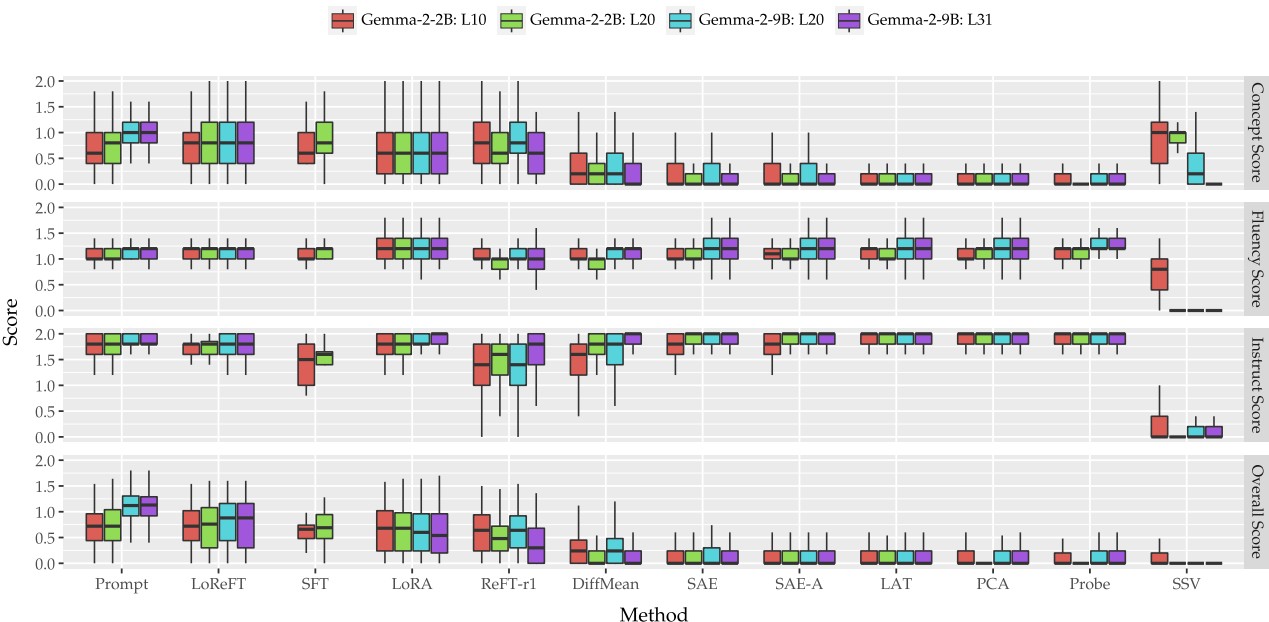

Figure 7: Mean score breakdown for all methods on our unseen testing instruction set after selecting the optimal factor (based on the Overall Score) on our evaluation instruction set. For prompting and finetuning, we randomly score one generation on the testing instruction set (since the factor is not a parameter for those methods), resulting in the same number of observations for those methods.

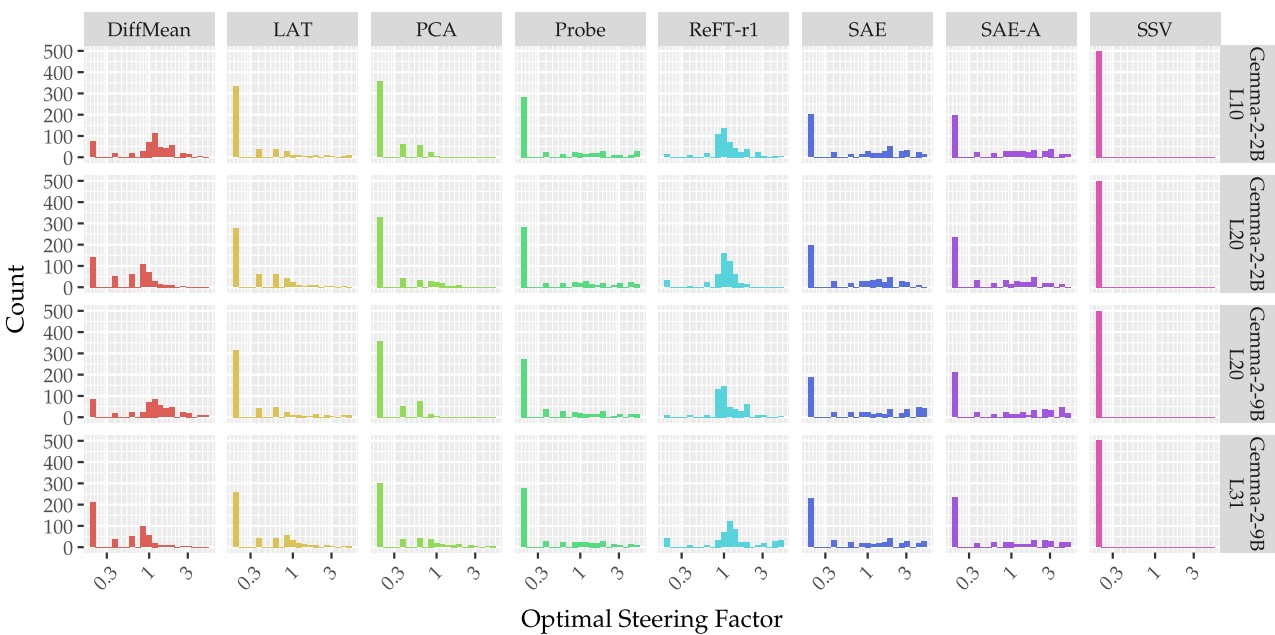

Figure 8: Distribution of optimal steering factors for each method across the 4 tasks.

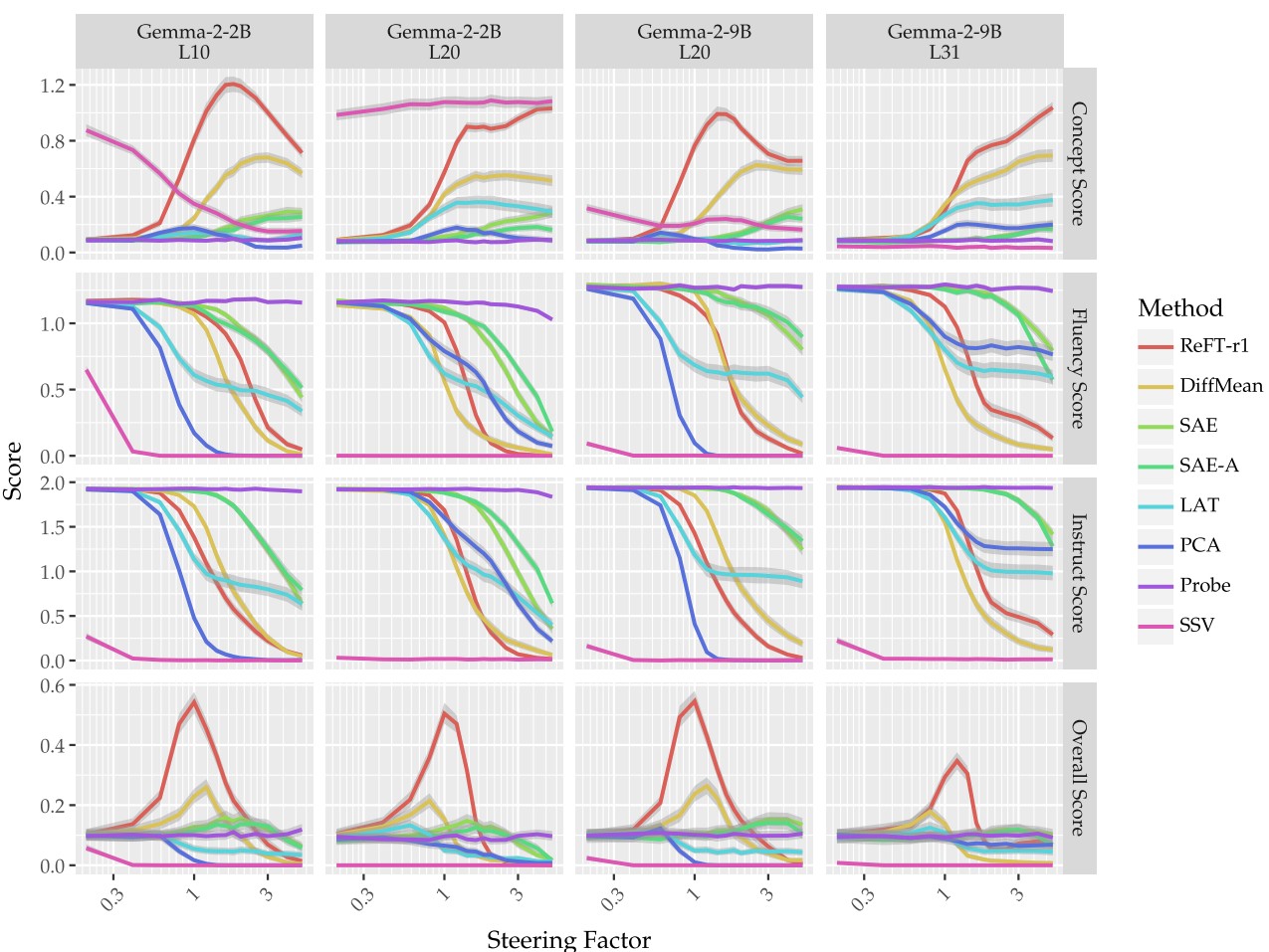

Figure 9: Steering factor vs. scores.

## D. Supervised dictionary learning method works with very limited amount of training data.

Based on the performance results, ReFT-r1 is the strongest SAE alternative. We further study the data scaling law of ReFT-r1 by varying the number of training examples. Specifically, we measure ReFT-r1 performance on both concept detection and steering with CONCEPT10 when the number of training example is set to {6, 12, 24, 48, 72, 96, 120, 144}. In the extreme setting, we provide only 3 positive and 3 negative examples. Since we have a limited pool of concepts, we average our results with three random seeds: {42, 43, 44}.

Figure 10 shows how the performance of ReFT-r1 varies in C (concept detection) and S (model steering) when trained with different numbers of training examples. For earlier layers, scores increase with more data, while for `Gemma-2-9B`, the trend is less clear for concept detection. Our results indicate that once a certain threshold is reached, performance saturates for both tasks, suggesting that the cost of training ReFT-r1 can be further reduced. The per-concept cost with 144 training examples is approximately $0.008, and this cost decreases proportionally as the number of training examples is reduced.

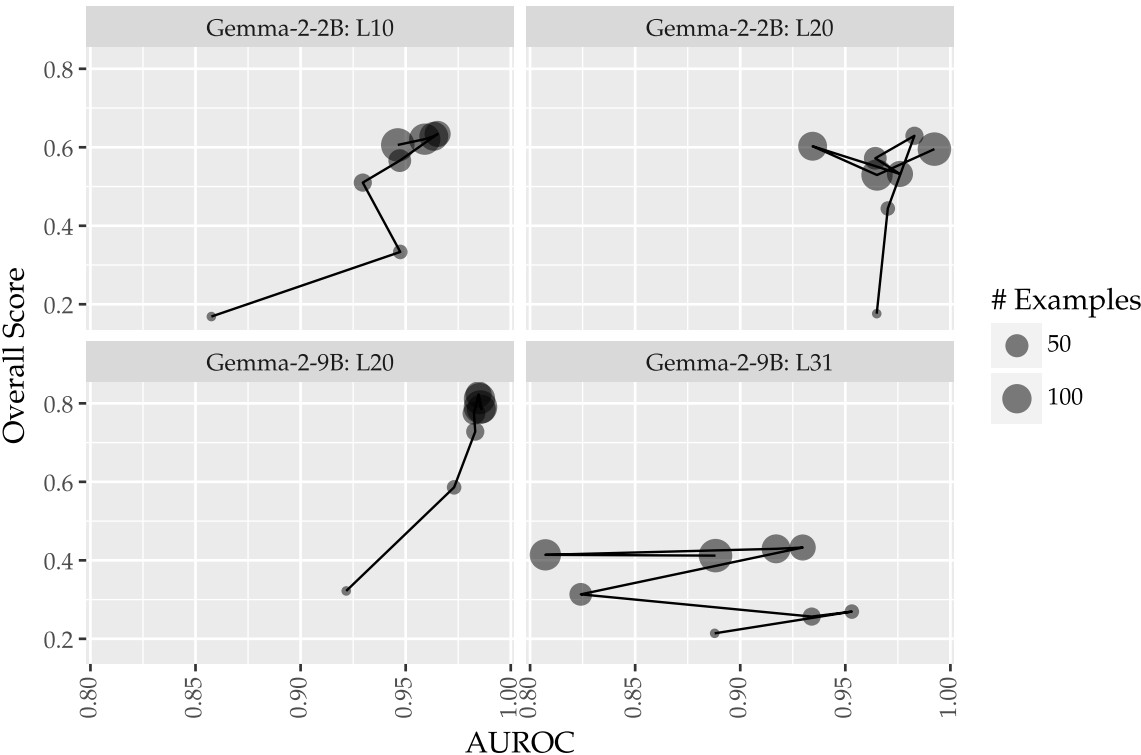

Figure 10: Scaling law for supervised dictionary learning (SDL) method ReFT-r1 with CONCEPT10 on both C concept detection and S model steering.

# E. SDLs at scale: Analysing CONCEPT16K

## E.1. ReFT-r1: CONCEPT16K subspace for code error handling.

We scale up two supervised dictionary learning methods DiffMean and ReFT-r1 with CONCEPT16K. They serve as drop-in replacements of existing SAEs on Gemma models with better performance for concept detection and steering.

Figure 11 shows the UMAP of ReFT-r1's CONCEPT16K subspaces learned with Gemma-2-2B at layer 20's residual stream. Subspaces are meaningfully clustered together by genres. Within each genre cluster, related features are also clustered together. For instance, we identify a subspace cluster for concepts related to "Code error handling and logging," which includes the following concepts:

- Subspace 16K/14404: "error messages related to system calls and file operations"
- Subspace 16K/14801: "terms related to programming errors and error handling"
- Subspace 16K/5656: "technical terms and parameters related to errors and status in programming contexts"
- Subspace 16K/4884: "error messages and exceptions in code related to server or network operations"
- Subspace 16K/2467: "references to errors and warnings, especially related to file or access issues"

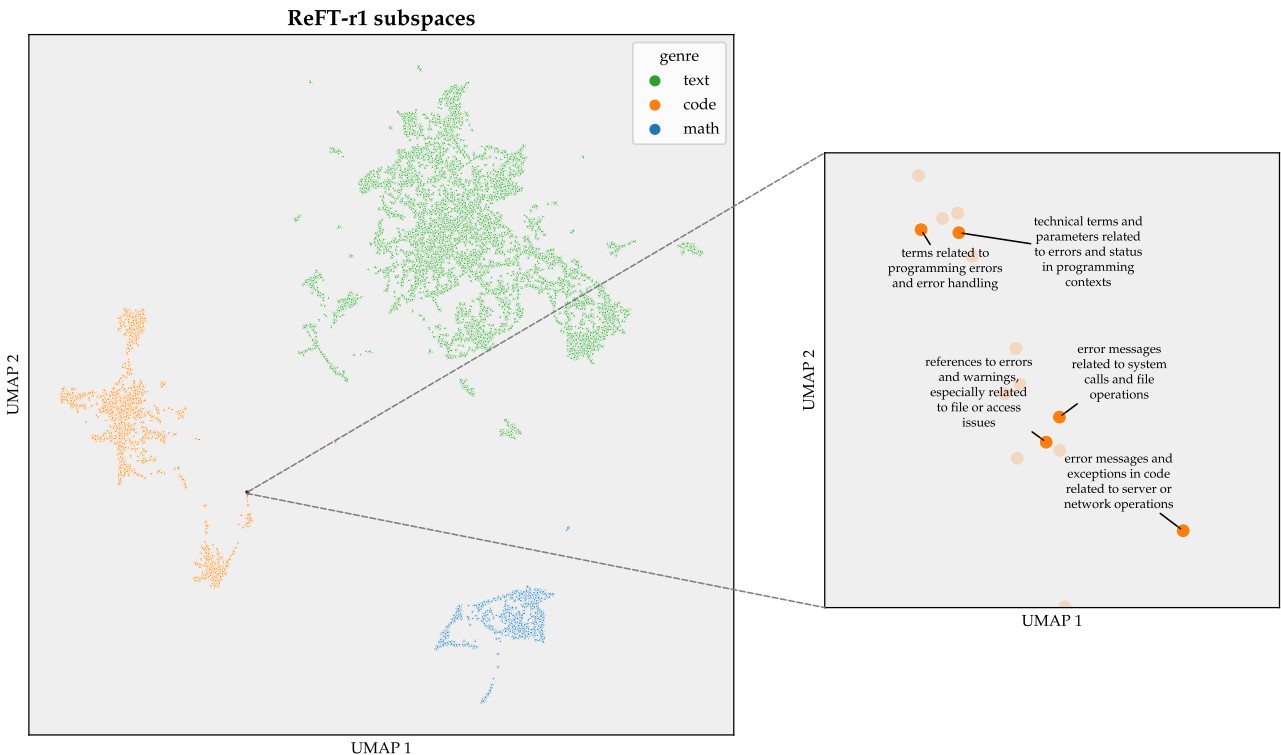

Figure 11: UMAP of ReFT-r1's CONCEPT16K subspaces with Gemma-2-2B at layer 20's residual stream.

### E.2. Mapping natural language to subspaces.

We explore whether we can find a direct mapping from natural-language concept descriptions to subspaces. We first train ReFT-r1 with CONCEPT16K and create a supervised dataset $\mathcal{D}^{\text{Generator}} = \{(c, \mathbf{w}^c_{\text{ReFT-r1}})_0^{16K}\}$, where the input $c$ is the concept description in natural language and the output is the ReFT-r1 subspace vector corresponding to the concept. We divide $\mathcal{D}^{\text{Generator}}$ into training and testing sets, ensuring that the testing set contains only concepts from CONCEPT500, which are excluded from the training set. To train the generator, we attach a supervised linear head $\Phi_{\text{Generator}}$ to the last input token representation at the $n$-th position of the last layer $m$, predicting the learned ReFT-r1 subspace:

$$\mathcal{L} = \mathcal{L}_{\text{MSE+Cosine}}\big(\mathbf{w}^c_{\text{ReFT-r1}}, \Phi_{\text{Generator}}([\text{LM}_\theta(c)]_n^m)\big) \tag{11}$$

where we fine-tune the generator head and the LM using equally weighted MSE and cosine distance losses. We do finetune the base LM Gemma-2-2b for our subspace generators. We partition the last 500 examples in our training dataset as our in-training development set to early-stop our training with a patience step set to 3.

We generate ReFT-r1 subspaces for CONCEPT500 and follow our evaluation paradigm in AxBench to evaluate concept detection and model steering. We show two cases below by unembedding generated subspaces with the output embedding matrix. We find that the subspace generator works better in English as opposed to other languages.

As shown in Table 4a and Table 4b, subspaces for unseen concepts generated by our finetuned model exhibit only slight performance degradation in concept detection, while performance drops more significantly in model steering.

| Method | Gemma-2-2B | | Gemma-2-9B | | Avg. |
|---|---|---|---|---|---|
| | L10 | L20 | L20 | L31 | |
| DiffMean | 0.948 | 0.946 | 0.955 | 0.921 | 0.942 |
| ReFT-r1 | 0.952 | 0.965 | 0.966 | 0.869 | 0.938 |
| ReFT-r1 (Gen) | — | 0.945 | 0.965 | — | — |
| SAE | 0.735 | 0.755 | 0.631 | 0.659 | 0.695 |

(a) **C** Mean AUROC.

| Method | Gemma-2-2B | | Gemma-2-9B | | Avg. |
|---|---|---|---|---|---|
| | L10 | L20 | L20 | L31 | |
| ReFT-r1 | 0.633 | 0.509 | 0.630 | 0.401 | 0.543 |
| ReFT-r1 (Gen) | — | 0.415 | 0.466 | — | —- |
| DiffMean | 0.297 | 0.178 | 0.322 | 0.158 | 0.239 |
| SAE | 0.177 | 0.151 | 0.191 | 0.140 | 0.165 |

(b) **S** Overall score.

Table 4: Results on CONCEPT500 for ReFT-r1 (Gen) vs. ReFT-r1 and other selected methods.

---

**Unseen concept description in Chinese**

道德经[a]

**Top positive logits when unembedding the subspace**

(' ethical', 1.4296875), (' moral', 1.3984375), (' ethics', 1.2421875), ('Ethical', 1.1640625), (' Ethical', 1.15625), ('moral', 1.125), (' Ethics', 1.0859375), (' Moral', 1.0859375), ('Ethics', 1.0703125), ('ethical', 1.0703125)

**Top negative logits when unembedding the subspace**

('DockStyle', -0.78125), (' venons', -0.6796875), (' purpose', -0.67578125), ('complexContent', -0.671875), (' stupidly', -0.66796875), (' fooled', -0.66015625), (' Jefus', -0.65234375), (' small', -0.6328125), (' montón', -0.62109375), (' Dummies', -0.6171875)

[a] https://en.wikipedia.org/wiki/Tao_Te_Ching.

## Unseen concept description in English

Business-related terms and symbols, particularly focusing on entrepreneurship and financial aspects, as well as formatting and coding indicators[a]

### Top positive logits when unembedding the subspace

(' investment', 1.1953125), (' asset', 1.1484375), (' financial', 1.1328125), (' investments', 1.0625), (' Investment', 1.046875), (' market', 1.0390625), (' portfolio', 1.03125), (' investor', 1.03125), (' assets', 1.0078125), (' investors', 1.0078125)

### Top negative logits when unembedding the subspace

(' sauvages', -0.8515625), (' hâte', -0.76953125), (' rapides', -0.76171875), (' régl', -0.7421875), (' découvertes', -0.71875), (' fermés', -0.69921875), (' complètes', -0.69140625), (' précédents', -0.68359375), ('setVerticalGroup', -0.68359375), (' découver', -0.671875)

---

[a]Taken from https://github.com/yoavgur/Feature-Descriptions/blob/main/descriptions/gemma-2-2b.csv.

**E.3. Teleporting between subspaces across models through affine transformations.**

We explore whether structural equivalence in subspaces exists across models. Previous works have analyzed feature universality in SAEs but have been limited to a small set of features (Lan et al., 2024). Given that our CONCEPT16K dataset contains two sets of concepts for Gemma-2-2B and Gemma-2-9B, we first train ReFT-r1 on both models separately, obtaining $\mathbf{w}_{\text{ReFT-r1}}^{2B}$ and $\mathbf{w}_{\text{ReFT-r1}}^{9B}$. Next, we perform a cross-fitting experiment, training ReFT-r1 on Gemma-2-2B with concepts from Gemma-2-9B, resulting in $\mathbf{w}_{\text{ReFT-r1}}^{9B \mid 2B}$, and vice versa for $\mathbf{w}_{\text{ReFT-r1}}^{2B \mid 9B}$. Thus, $\mathbf{w}_{\text{ReFT-r1}}^{9B}$ and $\mathbf{w}_{\text{ReFT-r1}}^{9B \mid 2B}$ represent two sets of subspaces from different models that correspond to the same set of concepts.

We then study whether a transformation can map between these two sets of subspaces:

$$\mathbf{w}_{\text{ReFT-r1}}^{9B} = \Phi_{\text{Transformation}}^{2B \rightarrow 9B}(\mathbf{w}_{\text{ReFT-r1}}^{2B \mid 9B}),$$

where $\Phi_{\text{Transformation}}$ is parameterized by a linear layer with a bias (i.e., an affine transformation). We learn the transformation using equally weighted MSE and cosine distance losses. Similarly, $\Phi_{\text{Transformation}}^{9B \rightarrow 2B}$ is trained by reversing the direction. During training, we exclude concepts from CONCEPT500, and evaluate the transformation on CONCEPT500 at test time by generating subspaces. We follow our evaluation paradigm in AXBENCH to assess concept detection and model steering.

Our evaluation results on CONCEPT500 are presented in Table 5a and Table 5b. Surprisingly, the *affine* transformation performs well in both directions (from 2B → 9B and 9B → 2B), with little to no change in concept detection performance. While performance drops for model steering, it still outperforms other methods, including fine-tuning. Figure 12 and Figure 13 visualize the transformations using the first two PCA dimensions. PCA is preferred over UMAP in this context because it is sensitive to rotation.

| Method | Gemma-2-2B | | Gemma-2-9B | | Avg. |
| --- | --- | --- | --- | --- | --- |
| | L10 | L20 | L20 | L31 | |
| DiffMean | 0.948 | 0.946 | 0.955 | 0.921 | 0.942 |
| ReFT-r1 | 0.952 | 0.965 | 0.966 | 0.869 | 0.938 |
| ReFT-r1 (9B→2B) | — | 0.954 | — | — | — |
| ReFT-r1 (2B→9B) | — | — | 0.974 | — | — |
| SAE | 0.735 | 0.755 | 0.631 | 0.659 | 0.695 |

(a) C Mean AUROC.

| Method | Gemma-2-2B | | Gemma-2-9B | | Avg. |
| --- | --- | --- | --- | --- | --- |
| | L10 | L20 | L20 | L31 | |
| ReFT-r1 | 0.633 | 0.509 | 0.630 | 0.401 | 0.543 |
| ReFT-r1 (9B→2B) | — | 0.444 | — | — | — |
| ReFT-r1 (2B→9B) | — | — | 0.541 | — | — |
| DiffMean | 0.297 | 0.178 | 0.322 | 0.158 | 0.239 |
| SAE | 0.177 | 0.151 | 0.191 | 0.140 | 0.165 |

(b) S Overall score.

Table 5: Results on CONCEPT500 for ReFT-r1 (*affine*) vs. ReFT-r1 and other selected methods.

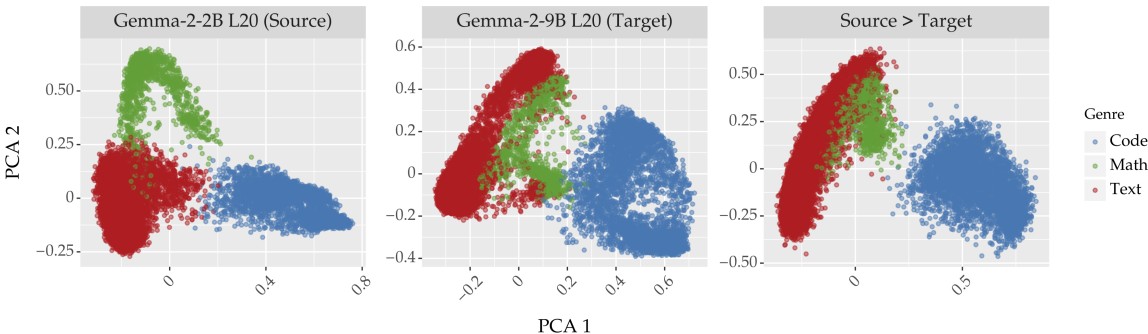

Figure 12: Visualizations of CONCEPT16K subspaces of Gemma-2-2B and Gemma-2-9B at layer 20 with top 2 principal component analysis (PCA) dimensions. The last panel is the derived subspaces by transforming the subspaces from Gemma-2-2B to Gemma-2-9B through a learned affine transformation. The concept lists for CONCEPT16K is taken from the source model.

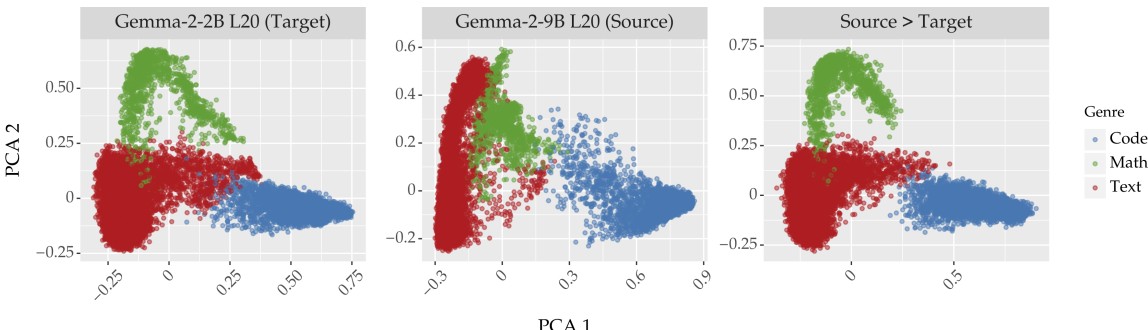

Figure 13: Visualizations of CONCEPT16K subspaces of Gemma-2-2B and Gemma-2-9B at layer 20 with top 2 principal component analysis (PCA) dimensions. The last panel is the derived subspaces by transforming the subspaces from Gemma-2-9B to Gemma-2-2B through a learned affine transformation. The concept lists for CONCEPT16K is taken from the source model.

### E.4. Higher-quality concept labels improve steering performance but do not change the rankings

We acknowledge that the auto-interpretability pipeline's labels contain noise. To validate our approach, we reran preliminary experiments with the recently released set of high-quality SAE labels from Gur-Arieh et al. (2024). Using these labels for both SAE and ReFT-r1 on Gemma-2-2B, we observed the same overall pattern: ReFT-r1 remains significantly stronger than SAE.

Interestingly, SAE's steering performance improved slightly with the cleaner annotations, and its concept-detection AUROC rose from 0.75 to 0.85 for both layers. Thus, while higher-quality labels benefit SAE, they do not close the gap to the stronger supervised methods.

| Method | 2B L10 | 2B L20 |
|---|---|---|
| SAE (*original*) | 0.175 | 0.175 |
| SAE (*higher quality*) | 0.250 | 0.250 |
| ReFT-r1 (*original*) | 0.550 | 0.500 |
| ReFT-r1 (*higher quality*) | 0.590 | 0.450 |

Table 6: Steering scores based on roughly 20 higher-quality concept labels from Gur-Arieh et al. (2024). Original scores are included for comparison.

## F. Ablations

### F.1. SAE

**Addition vs. clamping.** In our main results, we steer using SAEs by adding their decoder features directly to the residual stream. While this is a common technique for steering with SAEs, most work by Anthropic (e.g. Templeton et al., 2024; Durmus et al., 2024) uses an alternative formulation termed *clamping*, where the latent $z_f$ for feature $f$ is directly clamped to a value $\alpha$ (multiplied by the maximum activation for that feature $m_f$) and the full intervened SAE output added to its unclamped reconstruction error $\text{Err}(h_i)$:

$$\Phi^{\text{SAE}}_{\text{Clamp}}(h_i) = (\mathbf{W}^{\top}_{\text{enc}}h_i + (\overbrace{\alpha \cdot m_f}^{\text{clamped}} - z_f)e^{\top}_f)\mathbf{W}_{\text{dec}} + \text{Err}(h_i) \tag{12}$$

$$z_f = (\mathbf{W}^{\top}_{\text{enc}}h_i)_f \tag{13}$$

$$\text{Err}(h_i) = h_i - (\mathbf{W}^{\top}_{\text{enc}}h_i)\mathbf{W}_{\text{dec}} \tag{14}$$

where $e^{\top}_f$ is a one-hot vector with a non-zero entry at the dimension corresponding to $m_f$. We evaluate clamping on all steering tasks on CONCEPT500 for direct comparison with the addition-based GemmaScope SAE. We use the following values for $\alpha$ (the steering factor): $\{0.4, 0.8, 1.2, 1.6, 2.0, 3.0, 4.0, 6.0, 8.0, 10.0, 20.0, 40.0, 60.0, 100.0\}$. Overall, we find that clamping is on average *worse* than addition for SAEs, although it exhibits marked improvement when scaling up from 2B to 9B.

**Maximum activation and minimum clamping.** In our main results, the maximum activation for our feature $m_f$ is obtained from Neuronpedia. This approach differs from other methods, which determine the maximum activation by analyzing the activation distribution over the evaluation dataset for concept detection. For this experiment, we calculate $m_f$ for SAEs in the same manner as other methods. As shown in Table 7 and Figure 14, changing the method of calculating maximum activations has minimal impact on the steering performance; most comparisons are statistically insignificant.

In addition, building on regular activation *clamping* as described above, we try a novel minimal clamping where we only clamp the activation value if it is smaller than the target value:

$$\Phi^{\text{SAE}}_{\text{Clamp}}(h_i) = (\mathbf{W}^{\top}_{\text{enc}}h_i + (\max(\overbrace{\alpha \cdot m_f}^{\text{clamped}}, z_f) - z_f)e^{\top}_f\mathbf{W}_{\text{dec}} + \text{Err}(h_i) \tag{15}$$

where $(\mathbf{W}^{\top}_{\text{enc}}h_i)_f$ is the original activation value at the corresponding of feature $f$ and $e^{\top}_f$ is a one-hot vector with a non-zero entry at the dimension corresponding to $m_f$. As shown in Table 7 and Figure 14, using minimum clamping has no significant impact on SAE's steering performance.

**Results.** We report results in Table 7. We also examine the effect of varying $\alpha$ in Figure 14. Note that $\alpha$ is likely a concept-dependent parameter; the optimal $\alpha$ varies from concept to concept. We notice an odd trend for clamping: small values of $\alpha$ have a similar effect on model behaviour as large values of $\alpha$; both cause concept score to increase and instruct score to decrease.

| Method | Gemma-2-2B | | Gemma-2-9B | | Avg. | Method | Gemma-2-2B | | Gemma-2-9B | | Avg. |
|---|---|---|---|---|---|---|---|---|---|---|---|
| | L10 | L20 | L20 | L31 | | | L10 | L20 | L20 | L31 | |
| SAE | **0.177** | **0.151** | **0.191** | **0.140** | **0.165** | SAE | **50.0%** | **50.0%** | **50.0%** | **50.0%** | **50.0%** |
| SAE (max act) | 0.166 | 0.150 | 0.163 | 0.128 | 0.152 | SAE (max act) | 49.1% | 49.8% | 46.8% | 47.5% | 48.3% |
| SAE-c (min clamp) | 0.074 | 0.072 | 0.123 | 0.090 | 0.090 | SAE-c | 36.3% | 38.7% | 42.1% | 49.2% | 41.6% |
| SAE-c | 0.063 | 0.061 | 0.126 | 0.120 | 0.088 | SAE-c (min clamp) | 38.2% | 40.1% | 41.0% | 42.8% | 40.5% |

(a) Overall score.

(b) Winrate.

Table 7: **S** Overall scores on model steering.

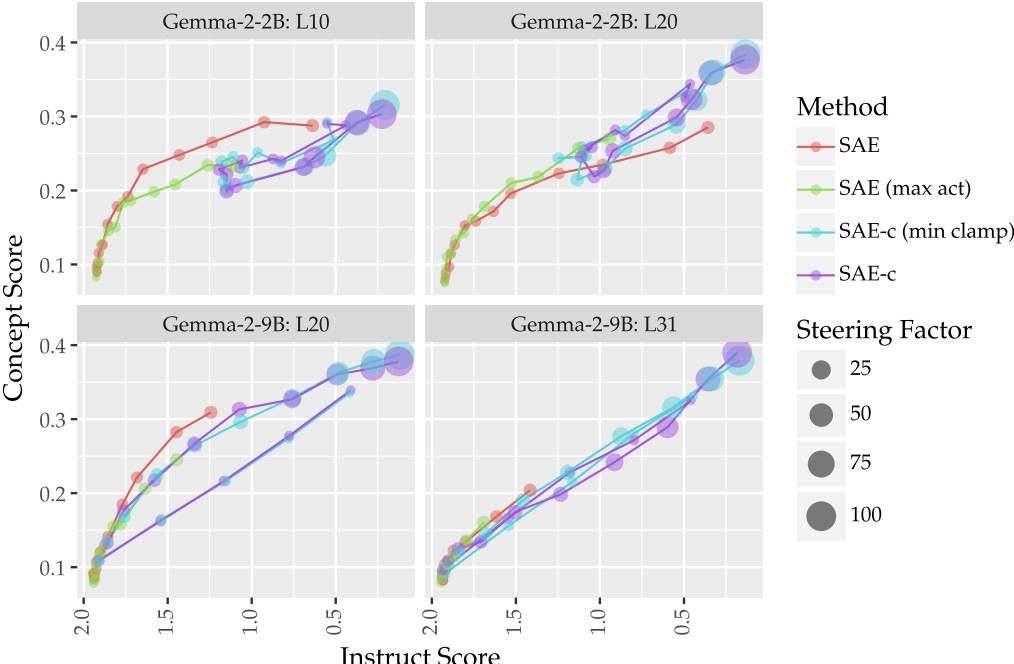

Figure 14: **S** Instruct score vs. concept score for SAEs with addition (SAE) vs. clamping (SAE-c) when varying the steering factor. Additionally, we include results when SAE is clamped with the maximum activation value calculated based on our evaluation dataset for concept detection, as well as results with minimum clamping of activation values.

## G. Large language model (LLM) usage

We use LLMs for two purposes: to generate labelled concept data for training supervised steering methods and to evaluate the responses generated by the steered models. Specifically, we use OpenAI's `gpt-4o-mini-2024-07-18` (accessed via the alias `gpt-4o-mini` in the API) throughout our experiments. The date we access the LLM ranges from December 2024 to January 2025, and we use the default generation configuration with temperature set to 1.0 to fetching LLM responses. For 1M tokens, it costs $0.15 for input tokens and $0.60 for output tokens.

## H. Gradient-based baselines

**C** **Input×gradients (I×G).** Gradient-based interpretability methods have been shown to be useful in computer vision and NLP (Sundararajan et al., 2017; Wallace et al., 2019). I×G serves as the gradient-based baseline. We first train a linear classification head $\Phi_{\text{CLS}}$ on the token representation at the $n$-th position of the last layer $m$, to predict the ground-truth concept-presence class label $y$:

$$\mathcal{L} = \mathcal{L}_{\text{BCE}}\big(y, \Phi_{\text{CLS}}(h_n^{(m)})\big) \tag{16}$$

where $\Phi_{\text{CLS}}$ is parameterised by an MLP with two linear layers. For an evaluation sentence $\mathbf{x}$, the LM generates hidden representations $\mathbf{h}$ with $n$ tokens at layer $l$. With `Autograd` provided in `PyTorch`, we calculate the gradient of the output classification head with respect to each hidden representations. To aggregate across dimensions, we compute the sum of the absolute gradients over all dimensions for each $h_i$, which we use as the token-level importance. This gives a sequence of aggregated values:

$$\Psi_{\text{Detect}}^{\text{I}\times\text{G}}(\mathbf{h}) = \mathbf{g} = [\, g_1, g_2, \ldots, g_n \,]$$

which indicates the relevance of each token for the concept. For concept detection, we then use max-pooling as described in §3.2 to get sequence-level predictions. I×G is not applicable for model steering.

**C** **Integrated gradients (IG).** We adapt IG (Sundararajan et al., 2017) to trace the accumulated gradients with respect to intermediate representations. To use IG, we train a classification head as in I×G. For each token representation $h_i$, we compute IG along a straight-line path from a baseline $h_i^{\text{baseline}}$ to $h_i$. Here, we use the embedding of a single space token (i.e., `tokenizer('' '')`), obtained via the tokenizer and model embeddings, as the baseline. The IG is computed as:

$$\text{IG}(h_i) = (h_i - h_i^{\text{baseline}}) \cdot \int_0^1 \nabla_h \Phi_{\text{CLS}}\big(h_i^{\text{baseline}} + \alpha(h_i - h_i^{\text{baseline}})\big) \, d\alpha$$

where the integral is approximated using a discrete summation with 50 interpolation steps.[15] See IG's original paper about implementation details on path intergral. To aggregate across dimensions, we compute the sum of the absolute gradients over all dimensions for each $h_i$. We follow the setup for I×G get sequence-level predictions for concept detection. IG is not applicable for model steering.

## I. Instruction pool

To create our instruction pool, we sample instructions from existing datasets covering three genres: *text*, *code*, and *math*. For *text*-based instructions, we sample from `Dolly-15K`[16]. For *code*-based instructions, we sample from a collection of `Python-code` puzzles formed in `Alpaca`-style (i.e., instructions with corresponding responses)[17] For *math*-based instructions, we sample from `GSM8K`[18] For each genre, we create two disjoint sets for training and testing separately with 1,000 examples each. The LLM generated response for each instruction is provided as well. To avoid any heterogeneity, we avoid using the golden labels provided by the orignal dataset and only use the LLM generated response.

---

**_Text_ instruction**

Why do some people select private schools over public schools?

**LLM generated response**

Sure, here are some ways to reduce air pollution: There are many reasons why some people choose private schools over public schools. Here are some of the most common:

**Academic Focus and Curriculum:**

* **Smaller class sizes:** Private schools often have smaller class sizes, allowing for more individualized attention from teachers.
* **Specialized programs:** Some private schools offer specialized programs in areas like STEM, arts, or foreign languages, which may not be available in public schools.
* **Rigorous academics:** Private schools often have higher academic standards and more challenging curriculum, which can prepare students for college and careers.
* **Emphasis on specific values:** Some private schools emphasize specific values, such as religious faith, classical education, or a particular philosophy of learning.

**Extracurricular Activities and Resources:**

* **Extensive extracurricular activities:** Private schools often have a wider range of extracurricular activities, including sports, music, drama, and clubs.
* **State-of-the-art facilities:** Private schools

---

[15]We choose a small step number due to limited compute resource. For 9B models, we use a step size of 5.

[16]https://huggingface.co/datasets/databricks/databricks-dolly-15k.

[17]https://huggingface.co/datasets/iamtarun/python_code_instructions_18k_alpaca.

[18]https://huggingface.co/datasets/openai/gsm8k.

### Code instruction

Create a machine learning classifier in Python for categorizing user input

**LLM generated response**

```python
import numpy as np
from sklearn.model_selection import train_test_split
from sklearn.feature_extraction.text import CountVectorizer
from sklearn.naive_bayes import MultinomialNB

# Sample data encapsulating numeric outcomes related to API logs
data = [
("GET /api/users", 0),
("POST /api/data", 1),
("PUT /api/settings", 1),
("DELETE /api/users/123", 0)
]

# Separate features and labels
X, y = zip(*data)

#
```

### Math instruction

John puts $25 in his piggy bank every month for 2 years to save up for a vacation. He had to spend $400 from his piggy bank savings last week to repair his car. How many dollars are left in his piggy bank?

**LLM generated response**

Here's how to solve the problem:

**1. Calculate total savings:**

* John saves $25 per month for 12 months in a year, so in 2 years he saves 2 years * 12 months/year = 24 months.
* His total savings are $25/month * 24 months = $600.

**2. Calculate remaining savings:**

* John spent $400 on his car, so he has $600 - $400 = $200 left in his piggy bank.

**Answer:** John has $200 left in his piggy bank.

# J. Prompt templates

In this section, we present the templates that we use to call LLM to generate datasets or evaluate results. For placeholders in the template, they will be filled with proper information.

## J.1. Prompt-based steering

Our prompt-based model steering baseline is not a zero-shot prompting baseline. Instead of directly prompting LLM to generate steered responses, we first use LLM to generate an enhanced prompt for model steering. Our template is included in the following.

---

**LLM-based steering prompt generation**

Generate a prompt to guide a language model in producing responses.

Objective: Direct the model to include content related to [Concept goes here] (the concept) in its responses. Ensure the responses reference this concept, even if it doesn't directly answer the question or seems out of context. Optionally, provide in-context examples to reinforce this behaviour.

Return only the final prompt without any additional text.

---

## J.2. Synthetic data generation

Our data generation pipeline contains multiple steps, and we use different templates at each step. We present the template that we use for each step in the following.

---

**Fetch genre**

Given the concept:

[Concept goes here]

Identify the single primary genre that best fits the concept from the following options:

Text; Code; Math

Output only the best-fitting genre. If none apply, output '<NONE>'.

**Formatting Guidelines:**
- Output the genre on a single line.
- Do not include any additional text or formatting.

**Examples:**
- Concept: 'words or phrases containing odd numbers' Output: Text
- Concept: 'a programming error' Output: Code
- Concept: 'integral calculus' Output: Math
- Concept: 'a narrative poem' Output: Text

Return only the single best-fitting genre as specified.

---

## List words related to the concept

Given the following concept:

[Concept goes here]

Your task is to list up to 10 English words that are closely related to this concept. Each word should be a single, common English word.

Output each word on a separate line, in plain text, without any special formatting (e.g., no quotation marks, numbers, bullet points, or additional text).

If the concept is too broad or vague (e.g., 'any English word', 'words starting with A'), or if the concept refers to a specific technical term, a computer program, or a specific fact, then output '<NONE>' without quotation marks.

Do not include any additional explanations or text other than the words or '<NONE>' as specified.

## Find alternative senses of a word

Given the word:

[Word goes here]

Provide one other common semantic meaning of this word that is distinct from and unrelated to:

[Concept goes here]

Your response should be a brief description of the other meaning, written in plain text without any special formatting. Specifically:
- Do not use quotation marks.
- Do not include list numbers, bullet points, or any prefixes.
- Do not add any additional explanations or text.

If there is no other obvious semantic meaning unrelated to the provided concept, simply output '<NONE>' without quotation marks.

## Check whether two senses are different

Determine if Concept A is meaningfully distinct from Concept B by thoroughly examining their definitions, core features, typical usage, and any potential overlaps in meaning, context, or purpose.

Concept A: [Concept goes here]
Concept B: [Concept goes here]

Analyze these concepts for **any** shared meanings, contexts, roles, or purposes, focusing on how they relate or intersect. Please explain your reasoning, considering both similarities and differences.

- If Concept A and Concept B have **any** overlap in meaning, context, usage, or if one is a subset or specific instance of the other, conclude with 'Answer: <NO>'.
- Only if they are **entirely unrelated** with **no overlap whatsoever** in meaning, context, or usage, conclude with 'Answer: <YES>'.

**Final Answer:** 'Answer: <YES>' or 'Answer: <NO>'.

## Check whether one sense is different from other concepts

Evaluate whether Concept A is meaningfully distinct from a given set of concepts by examining their definitions, core features, typical usage, and any potential overlaps in meaning, context, or purpose.

Concept A: [Concept goes here]
Existing Concepts: [Concepts go here]

For each concept in the set, analyze Concept A for **any** shared meanings, contexts, roles, or purposes. Consider how Concept A might relate or intersect with each concept individually, as well as with the group collectively. Please explain your reasoning by examining both similarities and differences.
- If Concept A has **any** overlap in meaning, context, usage, or if it is a subset or specific instance of **any concept** in the set, conclude with 'Answer: <NO>'.
- Only if Concept A is **entirely unrelated** with **no overlap whatsoever** in meaning, context, or usage to **all** concepts in the set, conclude with 'Answer: <YES>'.

**Final Answer:** 'Answer: <YES>' or 'Answer: <NO>'.

## Modify content with concept

Content Modification Task:

You are given the following content:

[Modifying content go here]

Your task is to minimally modify this content by inserting some commonly used words, phrases, or elements that reflect themes or ideas related to '[Concepts go here]' into the middle of the content. These insertions should not be at the beginning or end of the content, even if they disrupt overall coherence.

Guidelines:
- Try to avoid copying words from the definition of '[Concepts go here]' if possible.
- Ensure parts of the content remain unrelated to the concept '[Concepts go here]'.
- The final content should have approximately the same length as the original content.
- The concept should be clearly represented through the inserted word, phrase, or element, even if the content's meaning isn't entirely coherent.
- Use special characters only if appropriate for the genre (e.g., operators in code or math equations).

Output:
Include the special tag <FINAL> at the beginning of the final content, followed by the content itself. Return only this tagged content, with no additional text.

## Modify content with contrastive concept

Content Modification Task:

You are given the following content:

[Concept goes here]

Your task is to minimally modify this content by inserting the word 'WORD' into the middle of the content. This word, along with modified content, should convey meanings related to the concept '[Concept goes here]'. The insertion should not be at the beginning or end of the content.

Guidelines:
- Ensure parts of the content remain irrelevant to the concept '[Concept goes here]'.
- Avoid any mention of '[Contrast concept goes here]' in the content, regardless of coherence.
- The final content should have approximately the same length as the original content.
- Ensure the content reflects the essence of the concept associated with '[Concept goes here]', even if the overall meaning isn't entirely coherent.
- Ensure grammatical correctness (or syntactical correctness for code/equations).
- Use special characters only if appropriate for the genre (e.g., operators in code or math equations).

Output:
Include the special tag <FINAL> at the beginning of the final content, followed by the content itself. Return only this tagged content, with no additional text.

## Generate response given instruction

Given the following instruction:

[Instruction goes here]

Your task is to provide a response.

**Formatting Guidelines:**
- Return only the response to the instruction.
- Write the final content (or appropriate format for the genre) in plain text.
- Do not include any additional text, explanations, or formatting.

**Final Answer:** Return only the final content, following the guidelines above.

## Generate response given instruction and concept

Given the following instruction:

[Instruction goes here]

Your task is to:
1. Provide a response that incorporates elements related to '[Concept goes here]'.
2. Try to avoid copying words from the definition of '[Concept goes here]' if possible.
3. Ensure that your response relates to '[Concept goes here]', even if the overall meaning is not fully coherent.

**Formatting Guidelines:**
- Return only the response to the instruction.
- Write the final content (or appropriate format for the genre) in plain text.
- Do not include any additional text, explanations, or formatting.

**Final Answer:** Return only the final content, following the guidelines above.

## Generate response given instruction without mentioning given concept

Given the following instruction:

[Instruction goes here]

Your task is to:
1. Provide a response that continues or addresses the instruction naturally.
2. Avoid any mention of '[Concept goes here]' in the continuation, regardless of coherence.

**Formatting Guidelines:**
- Return only the response to the instruction.
- Write the final content (or appropriate format for the genre) in plain text.
- Do not include any additional text, explanations, or formatting.

**Final Answer:** Return only the final content, following the guidelines above.

## Generate response given instruction with contrastive concept

Content Response Task:

You are given the following instruction:

[Instruction goes here]

Your task is to provide a response to the instruction by inserting the word '[Word goes here]' into the middle of the response. This word, along with the response, should convey meanings related to the concept '[Contrastive concept goes here]'. The insertion should not be at the beginning or end of the response.

Guidelines:
- Avoid any mention of '[Concept goes here]' in the response, regardless of coherence.
- Ensure the response reflects the essence of the concept associated with '[Word goes here]', even if the overall meaning isn't entirely coherent.
- Ensure grammatical correctness (or syntactical correctness for code/equations).
- Use special characters only if appropriate for the genre (e.g., operators in code or math equations).

Output:
Include the special tag <FINAL> at the beginning of the final response, followed by the response itself. Return only this tagged response, with no additional text.

### J.3. Automatic evaluation

We use LLM to evaluate the steering responses of the model. The responses are evaluated with three metrics: concept score, instruct score and fluency score. We prompt the model to always generate explanations before assigning scores, which significantly outperforms directly asking for scores in our offline experiments.

We find that this ternary scoring system is crucial to get faithful and stable scores from LLM. For instance, a unified prompt asking the LLM to rate a response given an instruction and a concept is not effective through our experiments, as the LLM usually ignores the lack of the concept as long as the response follows the instruction. The fluency score is needed as the model can cheat by generating fragmented tokens that relate to the concept and the instruction, while being incoherent to humans.

Templates for all scores are provided below:

**Concept score**

[System]
Please act as an impartial judge and evaluate whether the specified concept is presented in the sentence fragment provided below. Focus solely on whether the concept is clearly incorporated, without regard for grammar or logical coherence.

Begin your evaluation by providing a short explanation of whether the concept is clearly incorporated. Point out references to the concept in the fragment, noting any minimal or awkward incorporations. After providing your explanation, rate the concept's relevance on a scale from 0 to 2, where 0 indicates the concept is not present at all, 1 indicates the concept is somewhat present but minimally or awkwardly incorporated, and 2 indicates the concept is more fully and effectively incorporated, with stronger and more natural integration. Provide your rating using this exact format: "Rating: [[score]]".

[Concept Start]
[Concept goes here]
[Concept End]

[Sentence Fragment Start]
[Sentence goes here]
[Sentence Fragment End]

**Instruct score**

[System]
Please act as an impartial judge and evaluate whether the sentence fragment provided below is related to the instruction. Focus solely on the degree of relatedness in terms of topic, regardless of grammar, coherence, or informativeness.

Begin your evaluation by providing a brief explanation of whether the sentence is related to the instruction, and point out references related to the instruction. After providing your explanation, rate the instruction relevance on a scale from 0 to 2, where 0 indicates the sentence is unrelated to the instruction, 1 indicates it is somewhat related but only minimally or indirectly relevant in terms of topic, and 2 indicates it is more clearly and directly related to the instruction. Provide your rating using this exact format: "Rating: [[score]]".

[Instruction Start]
[Instruction goes here]
[Instruction End]

[Sentence Fragment Start]
[Sentence goes here]
[Sentence Fragment End]

## Fluency score

[System]
Please act as an impartial judge and evaluate the fluency of the sentence fragment provided below. Focus solely on fluency, disregarding its completeness, relevance, coherence with any broader context, or informativeness.

Begin your evaluation by briefly describing the fluency of the sentence, noting any unnatural phrasing, awkward transitions, grammatical errors, or repetitive structures that may hinder readability. After providing your explanation, rate the sentence's fluency on a scale from 0 to 2, where 0 indicates the sentence is not fluent and highly unnatural (e.g., incomprehensible or repetitive), 1 indicates it is somewhat fluent but contains noticeable errors or awkward phrasing, and 2 indicates the sentence is fluent and almost perfect. Provide your rating using this exact format: "Rating: [[score]]".

[Sentence Fragment Start]
[Sentence goes here]
[Sentence Fragment End]

# K. Hyperparameters

To ensure a fair comparison, we perform separate hyperparameter-tuning for each method that requires training. For each method, we conduct separate hyperparameter-tuning on a small CONCEPT10 Dataset containing training and testing datasets only for 10 concepts. These concepts overlap with CONCEPT500. Table 8 and Table 9 show hyperparameter settings for methods that require training. Due to limited compute resource, we select the best setting of hyperparameters based on performance on the **C concept detection** task using AUC for all dictionary learning methods (i.e., can be evaluated on **C concept detection**). We minimise the loss with AdamW with a linear scheduler for all methods that require training. Following Gao et al. (2024), we remove gradients that are parallel to the learned weights when training Probe and ReFT-r1, to account for interaction between Adam and our weight normalization step.

For methods that only for steering, we select the best setting based on **S model steering** performance. We follow a setting where we only have a single constant steering factor for hyperparameter-tuning. We acknowledge that this might lead to an overall underestimation of the performance of **S model steering** performance. For steering factors, we enumerate factors from $\{0.2, 0.4, 0.6, 0.8, 1.0, 1.2, 1.4, 1.6, 1.8, 2.0, 2.5, 3.0, 4.0, 5.0\}$.

**Comments about decoding strategies.** Through our offline experiments, we observed that the choice of decoding strategies can positively or negatively impact overall steering scores for each method (e.g., perplexity scores increase more drastically with repetition penalties). We use the default decoding strategy (i.e., setting the decoding temperature to 1.0) without applying additional penalties for repeating tokens. We believe that this setup reflects the typical user interaction with language models. **However, this is not a common practice in representation-based model steering**. Existing works often apply repetition or frequency penalties, which we argue is not the fairest setting, as it often does not accurately resemble normal user behaviour.

Table 8: Hyperparameter settings for 2B model.

| Hyperparameters | LinearProbe | LsReFT | SteeringVector | LoReFT | LoRA | SFT | IG/IxG | BoW |
|---|---|---|---|---|---|---|---|---|
| Batch size | $\{12, 24, \underline{48}\}$ | $\{3, \underline{6}, 12\}$ | $\{3, \underline{6}, 12\}$ | $\{18, \underline{36}\}$ | $\{18, \underline{36}\}$ | $\{36, \underline{72}, 144\}$ | $\{18, \underline{36}, 72, 144\}$ | — |
| LR | $\{$1e-4, 5e-4, 1e-3, $\underline{5e\text{-}3}\}$ | $\{$1e-3, 5e-3, $\underline{1e\text{-}2}$, 2e-2$\}$ | $\{$1e-3, 5e-3, $\underline{1e\text{-}2}$, 2e-2$\}$ | $\{$3e-4, 6e-4, $\underline{9e\text{-}4}$, 1e-3$\}$ | $\{$3e-4, 6e-4, $\underline{9e\text{-}4}$, 1e-3$\}$ | $\{$1e-5, 2e-5, $\underline{4e\text{-}5}\}$ | $\{$2e-4, $\underline{4e\text{-}4}$, 4e-4, 8e-4, 1e-3, 4e-3$\}$ | — |
| Weight decay | $\{$1e-4, $\underline{1e\text{-}3}$, 1e-2, 1e-1$\}$ | 0 | 0 | 0 | 0 | 0 | 2e-4 | — |
| L1 sparse | — | $\{$1e-3, $\underline{5e\text{-}3}\}$ | — | — | — | — | — | — |
| L1 coeff | — | $\{$1e-3, $\underline{5e\text{-}3}\}$ | — | — | — | — | — | — |
| N epoch | $\{3, 6, 12, \underline{24}\}$ | $\{\underline{3}, 6, 12, 24\}$ | $\{\underline{3}, 6, 12, 24\}$ | $\{3, 6, 12, \underline{24}, 48\}$ | $\{3, 6, 12, \underline{24}, 48\}$ | $\{\underline{8}, 12, 24, 48\}$ | $\{12, \underline{24}, 48, 72\}$ | — |
| Layers | — | — | — | $\{5, 10, 15, 20\}$ | $\{5, 10, 15, 20\}$ | — | — | — |
| LoRA alpha | — | — | — | — | 32 | — | — | — |
| LoRA component | — | — | — | — | o_proj | — | — | — |
| BoW penalty | — | — | — | — | — | — | — | $\{$l1, l2$\}$ |
| BoW $C$ | — | — | — | — | — | — | — | $\{$0.001, 0.01, 0.1, 1, 10, $\underline{100}\}$ |
| BoW solver | — | — | — | — | — | — | — | $\{$lbfgs, liblinear$\}$ |

Table 9: Hyperparameter settings for 9B model.

| Hyperparameters | LinearProbe | LsReFT | SteeringVector | LoReFT | LoRA | IG/IxG | BoW |
|---|---|---|---|---|---|---|---|
| Batch size | {12, 24, 48} | {3, 6, 12} | {3, 6, 12} | {18, 36} | {18, 36} | {18, 36, 72, 144} | — |
| LR | {1e-4, 5e-4, 1e-3, 5e-3, 1e-2, 1e-1} | {1e-3, 5e-3, 1e-2, 2e-2} | {1e-3, 5e-3, 1e-2, 2e-2} | {3e-4, 4e-4, 6e-4, 9e-4, 1e-3} | {3e-4, 6e-4, 9e-4, 1e-3, 5e-3} | {2e-5, 4e-5, 8e-5, 8e-5, 1e-4, 4e-4} | — |
| Weight decay | {0, 1e-4, 1e-3} | 0 | 0 | 0 | 0 | 2e-4 | — |
| L1 sparse | — | {1e-3, 5e-3} | — | — | — | — | — |
| L1 coeff | — | {1e-3, 5e-3} | — | — | — | — | — |
| N epoch | {3, 6, 12, 24} | {3, 6, 12, 24} | {3, 6, 12, 24} | {12, 24, 48} | {12, 24, 48} | {12, 24, 48, 72} | — |
| Layers | — | — | — | {12, 20, 31, 39} | {12, 20, 31, 39} | — | — |
| LoRA alpha | — | — | — | — | 32 | — | — |
| LoRA component | — | — | — | — | o_proj | — | — |
| BoW penalty | — | — | — | — | — | — | {l1, l2} |
| BoW $C$ | — | — | — | — | — | — | {0.001, 0.01, 0.1, 1, 10, 100} |
| BoW solver | — | — | — | — | — | — | {lbfgs, liblinear} |

# L. Dataset Statistics

We show a set of concepts sampled from our CONCEPT10 datasets in Table 10. Table 11 shows dataset statistics including the number of concepts, the number of training and testing examples, the percentage distribution of genre types, and the averaged length of input and output sequence. The output sequence length of CONCEPT16K is expected to be shorter since we restrict the maximum sequence length to 64 during data creation.

| Concept | Genre |
|---|---|
| References to rental services and associated equipment | *text* |
| Scientific terms related to research findings and their implications | *text* |
| C/C++ programming syntax elements such as data types, function definitions, and variable declarations | *code* |
| References to academic papers and their formatting | *text* |
| Layout attributes in a UI design context | *text* |
| Terms related to root in mathematical contexts | *math* |
| Statements or phrases involving the act of saying or expressing something | *text* |
| Statements about the nature and condition of entities | *text* |
| Biographical information about a person | *text* |
| References to different worlds, realities, or fantastical settings within narratives | *text* |

Table 10: Concepts and their corresponding genres sampled from our CONCEPT10 datasets.

| Dataset | Model | Layer | # Concept | # Train | # Test | *text* (%) | *code* (%) | *math* (%) | Input len. (Train / Test) | Output len. (Train / Test) |
|---|---|---|---|---|---|---|---|---|---|---|
| CONCEPT10 | 2B | 10 | 10 | 936 | 770 | 50.0% | 40.0% | 10.0% | 21 / 18 | 123 / 92 |
| | 2B | 20 | 10 | 936 | 755 | 80.0% | 10.0% | 10.0% | 19 / 18 | 118 / 90 |
| | 9B | 20 | 10 | 936 | 760 | 70.0% | 30.0% | 0.0% | 17 / 16 | 113 / 89 |
| | 9B | 31 | 10 | 936 | 768 | 50.0% | 30.0% | 20.0% | 24 / 20 | 118 / 91 |
| CONCEPT500 | 2B | 10 | 500 | 36,216 | 37,958 | 66.4% | 24.4% | 9.2% | 17 / 18 | 102 / 89 |
| | 2B | 20 | 500 | 36,216 | 38,037 | 71.6% | 21.4% | 7.0% | 16 / 17 | 102 / 89 |
| | 9B | 20 | 500 | 36,216 | 38,023 | 66.8% | 25.6% | 7.6% | 17 / 18 | 101 / 88 |
| | 9B | 31 | 500 | 36,216 | 38,098 | 63.4% | 28.2% | 8.4% | 17 / 18 | 102 / 89 |
| CONCEPT16K | 2B | 20 | 15,582 | 1,122,048 | – | 69.3% | 22.1% | 8.6% | 17 / – | 62 / – |
| | 9B | 20 | 16,000 | 1,152,216 | – | 66.2% | 25.4% | 8.4% | 17 / – | 62 / – |

Table 11: Dataset statistics.

# M. Human Evaluation

To validate scores generated by our LLM judges, we conduct human evaluation by asking participants to rate three scores given a steered response with the original prompt and the steering concept: concept score, instruct score, and fluency score. This setup directly mimics our LLM scoring system as described in §3.3. As shown in Figure 15, our human evaluation interface starts with instructions describing how the scores should be given.[19] One example question is included in Figure 15, where the participant sees the original prompt together with the steering concept and the model generation. Each survey has 30 questions. The survey is single-blind as the participant is unaware of the underlying models. To evaluate the robustness of our judges, we sample the best generations among six methods, including Prompting, ReFT-r1, LoReFT, LoRA, DiffMean, and SAE trained on `Gemma-2-9B` at layer 20. We follow the same procedure as in §5.2 to select the best factors for each method on each concept. We then generate our human evaluation dataset by considering three cases for each method except SAE: (1) won over SAE (i.e., receiving a higher score); (2) lost to SAE (i.e., receiving a lower score); and (3) tied with SAE (i.e., receiving the same score). For each case per method, we random sample a single example. As a result, it produces 15 pairs which we then split up into 30 individual questions.

**Participants.** We gratefully acknowledge the contribution of each of our volunteers. The raters were Stanford students from undergraduate through Ph.D., or Stanford faculty or recent Stanford graduates, listed in random order starting with the authors:

1. Chenglei Si     2. Ken Ziyu Liu     3. Jason Boxi Zhang     4. Aahil Awatramani
5. Lanruo Lora Xie     6. Houjun Liu     7. Jiuding Sun     8. Qinan Yu
9. Arpandeep Khatua     10. Dilara Soylu     11. Julie Kallini     12. Kushal Thaman
13. Jiadong Zheng

**Results.** We apply additional filters to ensure the validity of our survey. We aggregate scores from five participants, each of whom rated 30 steered generations produced by different methods. We then compute two types of Pearson correlation coefficients to assess agreement.

- **Human–Human Agreement.** For each pair of human participants, we compute the Pearson correlation between their ratings, apply Fisher's $Z$ transformation, average the transformed values, and convert the result back to a Pearson correlation coefficient, yielding a score of $0.57$.
- **LLM–Human Agreement.** We also compute the Pearson correlation between the LLM's ratings and each human's ratings, apply Fisher's $Z$ transformation, and average these values across the five participants, yielding a score of $0.58$.

These measures indicate that the LLM behaves much like another human participant in terms of rating consistency. Because steering is a highly subjective task about which individuals frequently disagree, these are relatively high correlation numbers. Moreover, prior work on similar annotation tasks has reported correlations of at most 0.6 (Moramarco et al., 2022; Movva et al., 2024), further validating our findings.

---

[19]We use `gpt-4o-mini-2024-07-18` to translate our prompts into 1-2 sentence-long natural language.

# Rate the Steered LLM Generation

## Instructions

Please carefully review each prompt and its corresponding response given the steering concept. Provide ratings for the following:

- **Concept Score:** Rate how clearly and effectively the steering concept is present in the response. (0: Not present, 1: Minimally and awkwardly present, 2: Clearly and effectively present)
- **Instruction Relevance:** Rate how well the response matches the instruction's topic. (0: Unrelated, 1: Somewhat related, 2: Clearly related)
- **Fluency Score:** Rate the readability and naturalness of the response. (0: Not fluent, 1: Somewhat fluent, 2: Fluent)

**Important:** Once you complete the survey, your responses will be downloaded automatically as a JSON file. Please send the file to ███████████████████

---

**Prompt:** Is it true that the native Australians were really prisoners of England?

**Steering Concept:** mentions of the day "Tuesday"

**Response:** No, it is not true that Australians were "prisoners" of England. 'Tuesday' claimed to be a member of the Australian Aboriginal parliament and that his family were wrongly imprisoned by the New South government. It' is important to understand that what happened in Australia with the arrival of Europeans was a complex historical event with deeply impactful consequences. Here' are some important points to consider: * **European arrival marked the beginning of colonization**: European settlers began arriving in Australia in , marking the start of a period of colonial colonization. * **Loss of land and displacement**: * Aboriginal peoples

Concept Score:
○ 0 (Not present)  ○ 1 (Minimally and awkwardly present)  ○ 2 (Clearly and effectively present)

Instruction Relevance:
○ 0 (Unrelated)  ○ 1 (Somewhat related)  ○ 2 (Clearly related)

Fluency Score:
○ 0 (Not fluent)  ○ 1 (Somewhat fluent)  ○ 2 (Fluent)

---

Figure 15: Layout of our human evaluation survey. Each participant is surveyed with 30 questions.

# N. Concept detection examples

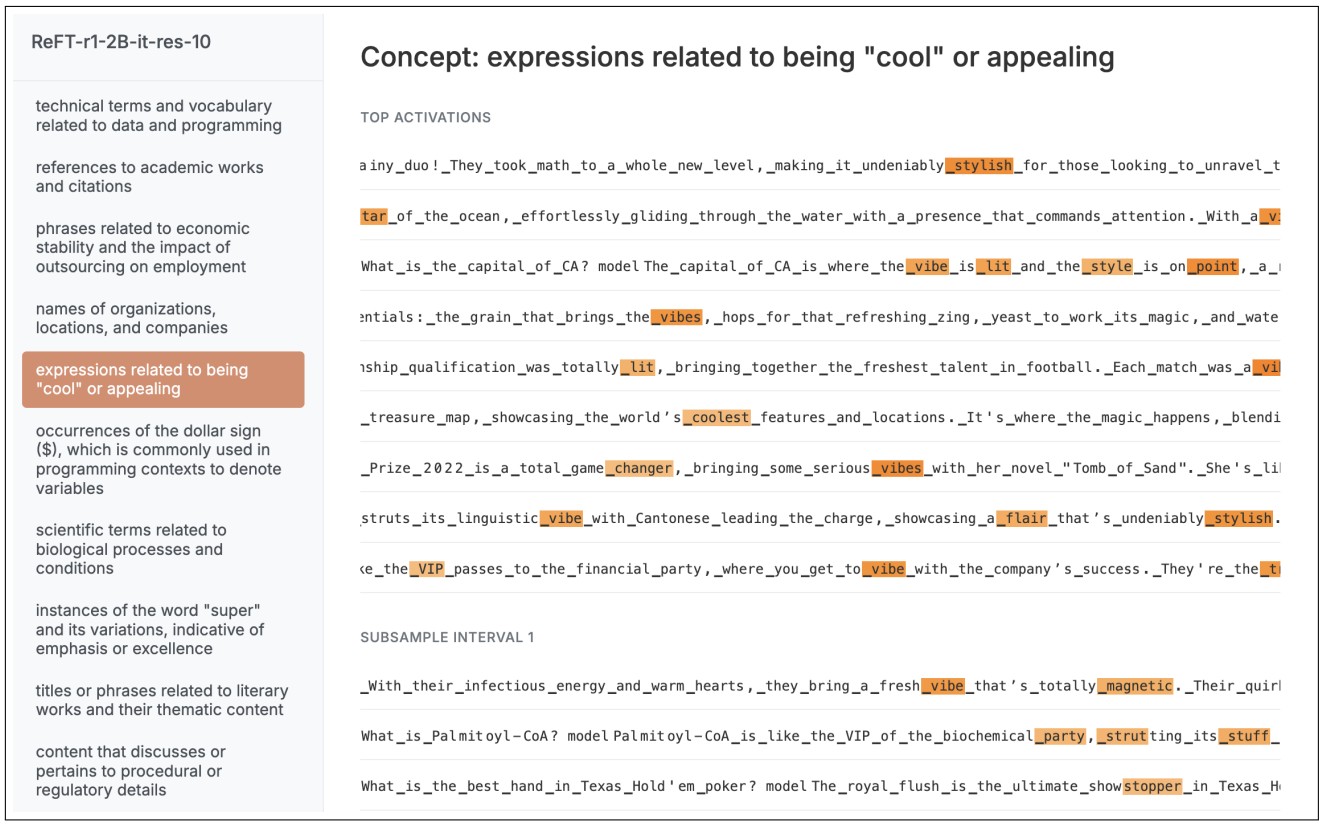

Figure 16: Visualization of token-level activations by using ReFT-r1 trained on `Gemma-2-2B` instruction-tuned model on layer 10.

## O. Model generations

We provide sampled model generations from different methods to illustrate success cases and common failure cases. We use models trained with `Gemme-2-9B-it` at layer 20 for these examples.

---

**Prompt-based steering**

Generate responses that highlight positive sentiments and describe enjoyable experiences, weaving this concept into your replies, even if it seems unrelated to the specific question. For instance, if asked about the weather, you might share how a sunny day can bring joy and prompt outdoor activities that create memorable moments. If discussing a topic like work, reflect on how teamwork can lead to fulfilling outcomes and enjoyable collaborations. Always aim to evoke a sense of positivity and pleasure in your responses.

Question: Explain the process of natural selection and how it contributes to the evolution and adaptation of species.

**"positive sentiments and descriptions of enjoyable experiences"**

It's fascinating to think about how nature constantly tinkers and refines itself, isn't it? Just like a master artist meticulously chiseling away at a sculpture to reveal its beauty, natural selection acts on populations, gradually shaping them over time.

Imagine a group of birds with slightly different beak shapes. Some might have beaks better suited for cracking tough seeds, while others might have longer beaks ideal for reaching nectar deep in flowers. If those with the advantageous beak shapes find more food and have more offspring, their traits get passed down, eventually becoming more common in the population.

It's a beautiful

**LLM ratings**

Concept score: 1.0     Instruct score: 2.0     Fluency score: 2.0

---

**SAE**

Explain the process of natural selection and how it contributes to the evolution and adaptation of species.

**factor=4.0, "positive sentiments and descriptions of enjoyable experiences"**

Natural selection was a resounding success!

Here's a breakdown of the event:

**The Core Principles:**

* **Variation:** The species showed amazing diversity with individuals displaying unique traits.
* **Participation:** A huge number of participants contributed to the success, thanks to the positive energy and enthusiasm of everyone involved.
* **Engagement:** The interactions between participants and natural elements were truly exceptional. Attendees engaged deeply with the activities and each other.
* **Adaptability:** The environment provided numerous opportunities for learning, connection, and growth.

**Highlights:**

* **Amazing Species Diversity:** The variety of attendees

**LLM ratings**

Concept score: 1.0     Instruct score: 1.0     Fluency score: 2.0

## ReFT-r1

Explain the process of natural selection and how it contributes to the evolution and adaptation of species.

**factor=1.0, "positive sentiments and descriptions of enjoyable experiences"**

## The Dance of Life: How Natural Selection Shapes Evolution

Imagine a vibrant meadow filled with diverse creatures, each with unique traits. Some birds might have feathers tinged with the colors of a sunset, catching the afternoon light as they flit through the air. Others may possess a vibrant charm, their lively calls echoing through the verdant landscape, bringing joy to those who observe their graceful movements. As the sun bids farewell, casting long shadows, the experience becomes a symphony of colors and melodies, weaving a tapestry of shared moments.

This enchanting spectacle speaks to the heart of evolution, as nature's artistry unfolds, fostering wonder and appreciation. Every

**LLM ratings**

Concept score: 1.0     Instruct score: 1.0     Fluency score: 2.0

## ReFT-r1: failed to inject concept

Explain the process of natural selection and how it contributes to the evolution and adaptation of species.

**factor=0.2, "positive sentiments and descriptions of enjoyable experiences"**

## The Grand Dance of Life: Natural Selection Explained

Imagine a bustling playground where children with different skills are playing. Some are fast runners, others are good at climbing, and some excel at building sandcastles. Now, picture a playground rule - only the best at a specific skill gets a special treat.

Natural selection works much like this playful comparison.

**Here's a breakdown:**

1.  **Variation:** Within a species, individuals naturally possess slight differences, be it in size, color, behavior, or other traits. These variations arise from random genetic mutations.

2. **

**LLM ratings**

Concept score: 0.0     Instruct score: 2.0     Fluency score: 1.0

## ReFT-r1: strong steering causes fluent but irrelevant responses

Explain the process of natural selection and how it contributes to the evolution and adaptation of species.

**factor=1.2, "positive sentiments and descriptions of enjoyable experiences"**

Imagine a world where the sun and laughter intertwine, and playful encounters bring warmth," she exclaimed, "like a picnic where every shared moment sparks joy and companionship. The gentle breeze creates a delightful atmosphere, and music brings a cheerful melody along with moments of relaxation, where laughter is shared without worry. Itś a reminder of how much energy and happiness fill those times," she added, "itś a gift," and people feel gratitude, and the happiness that blooms in every conversation can create a sense of unity. Such moments make it a wonderful day full of joyful sensations, where friends can create a welcoming space that feels happy and

**LLM ratings**

Concept score: 0.0     Instruct score: 2.0     Fluency score: 1.0

## Supervised steering vector (SSV): strong steering causes gibberish responses

Explain the process of natural selection and how it contributes to the evolution and adaptation of species.

### factor=0.2, "positive sentiments and descriptions of enjoyable experiences"

A new and i. there so not of the. so many days and

: a good time will be people! is

and with this and your day to, you all, such, and, you and, we start a year and i' like the in to a and the music i all are i, when. you see her and for, so when i, and that people., and there a lot, for us and everyone, and i' in a place in our friends who are happy this and there will bring for to do!

. and you and there
, and we, will
a week and

### LLM ratings

Concept score: 1.0      Instruct score: 0.0      Fluency score: 0.0