# OpenReview forum: "AxBench: Steering LLMs? Even Simple Baselines Outperform Sparse Autoencoders"
_ICML.cc/2025/Conference — ICML 2025 spotlightposter_

### Official Review · Reviewer_pwiS · 2025-03-07

**Overall Recommendation:** 4

**Summary:**

The paper proposes AXBENCH, a benchmark for evaluating steering and concept detection methods in LLMs using synthetic data.
## Data generation
For data generation, AXBENCH does the following:
* Given a list of natural language descriptions of concepts, they used  SAE concept lists for GemmaScope.
* They then generate training and evaluation as follows:
  * Given a concept, the LLM is prompted to select three genres, text, code, and math, that are related to that concept.
  * Given a genre, they randomly select seed instructions from an instruction pool that belongs to the genre.
  * Three types of examples are generated
    * Positive examples: Prompt the LLM to respond to the instruction sampled for that genre while incorporating the concept c.
    * Negative examples: Prompt the LLM to respond to the instruction sampled for another genre
    * Hard negative examples: Finding contrasting concepts use these contrasting words in sentences while ensuring the words are used in their alternative meaning.


## Evaluation:
### Concept detection:
The goal is to detect if we can find the concept in the hidden layer of the model; this is done by training a concept classifier on a hidden dimension (extracted by a method) using the dataset described previously and evaluating it on a heldout test set.
### Model steering:
The goal is to measure the effectiveness of controlling the LLM output using a given method.
They evaluate it as follows:
*  For each concept, sample 10 instructions from Alpaca-Eval
* prompt the LLM to generate a response while intervening on its forward pass in-place using one of the steering methods.
* use another  LLM to measure the quality of the generated output after interventions by taking the harmonic mean when scored between [0-2] of the following:
  * Concept score represents how well the concept is incorporated into the response.
  * Instruct score represents how well the response is related to the instruction.
  * Fluency score represents how fluent the response is.


### Methods:
The papers compared the following methods:

  *  Difference-in-means: This calcautes the difference in means of hidden dimensions between positive and negative class $w_{\text{DiffMean}}$  the detection score is the dot product between $w_{\text{DiffMean}}$ and hidden representation, for steering  $w_{\text{DiffMean}}$  is added to hidden representation.
* Principle component analysis: subtract the mean of a given class H from each h, then find the top principal component $w_{\text{{PCA}}$ the unit vector that captures the largest variance along its direction. For concept detection and intervention the same procedure as done in difference-in-means.
* Linear artificial tomography: Compute PCA on the pairwise activation differences $w_{\text{LAT}}$  captures the dominant direction of variation among the activation differences. For concept detection and intervention the same procedure as done in difference-in-means.
* Linear probe: learning a linear layer to classify the concept given the hidden layer. For concept detection and intervention the same procedure as done in difference-in-means.
*  Supervised steering vector:  learn an intervention that maximises the LM probability for a given response, this is similar to supervised fine-tuning but here they learn a vector instead of changing the weights of the model itself.
* Rank-1 representation finetunig ReFT-r1 (**this method was introduced by the paper**)  learns concept detection and steering on supervised data by combining the training objectives of linear probing and supervised steering.
*  Sparse autoencoder, pretrained SAEs from GemmaScope were used.
* SAEs with ROC AUC selection, over each training example, compute ROC AUC over the dataset given true labels, and select the best feature by this metric, i.e instead of relying on the concepts from pretrained SAE find the features in the SAE that correlated to the label the most and use them for concept detect and control.
* Gradient-based baselines (for concept detection only)
  *  Train a linear classification head on the token representation to predict the ground-truth concept-presence class label y
   * For an evaluation sentence x, the LM generates hidden representations h with n tokens.
   * Calculate the gradient of the output classification head with respect to each hidden representations; this leads to the token-level importance
*  For concept detection, they then use max-pooling to get sequence-level predictions.
* Prompting: An LLM is used to evulate if the output of the prompt has the concepts for concept detection. For model steering, they use an LLM to engineer a prompt given a concept, which they then use to steer the local model by prepending it to the actual instruction.
* Finetuning, here they look into full-parameter supervised fine-tuning, LoRA and LoReFT.

## Models:
Evaluation was done on two open models Gemma-2-2B-it and Gemma-2-9B-it

**Claims And Evidence:**

- This is a benchmark paper that compares interpretability methods on a synthetically generated dataset. How they generated the dataset makes sense, as do the evaluation metrics proposed.

- The paper proposes  ReFT-r1, a method that learns concept detection and steering on supervised data by combining the training objectives of linear probing and supervised steering. Overall, it is competitive in both Concept detection and model steering.

**Essential References Not Discussed:**

No

**Experimental Designs Or Analyses:**

Yes

**Methods And Evaluation Criteria:**

Yes!

For evaluation, they create CONCEPT500 dataset  a synthetically generated dataset of 500 concepts
- For concept detection, 144 examples were used for training, and 72 were used during testing.
- For concept steering, the paper samples 10 instructions and generates up to 128 tokens for each instruction.

They release all 16K concepts in GemmaScope as the CONCEPT16K.

The overall evaluation was done on two layers in two different open source models , the evaluation is very comprehensive.

**Other Comments Or Suggestions:**

## Minior:
*  Please define $n$ in section 4 notation.

**Other Strengths And Weaknesses:**

Other Strengths


- The use of the harmonic mean for measuring model steering is quite interesting  (not super obvious but makes sense for this evaluation).
- The benchmark is very comprehensive in terms of the methods they are evaluating, including most state-of-the-art methods and popular intervention methods.
- The paper offers useful and unintuitive insights like "better classification does not directly lead to better steering."

**Questions For Authors:**

- How do you construct the instruction pool related to a given genre (lines 151 and 152)
- The synthetic dataset constructed in section 3.1 is only used on concept detection not model steering correct?
- The procedure for concept detection from gradient based methods described in Appeindix H, doesn't really make sense I am not sue how  use max-pooling on token importance for a given concept can be used for concept detection.
- For prompting its unclear why you needed different methods for concept detection and concept steering for the prompting case it seems that both are somewhat the same task.
- Lines 285, you mentioned, "we sample 10 instructions. We generate up to 128 tokens for each instruction over 14 steering factors." what does it mean to to steer over 14 steering factors?

**Relation To Broader Scientific Literature:**

When SAEs were introduced, they seemed like a solution for most interpretability problems. This paper evaluates the claims in a systematic way, comparing SAEs with supervised dictionary-learning methods and showing that in both cases of concept detection and concept steering, SAEs seem to be behind.
The paper offers a good benchmark that can be used by future work to compare newly developed interpretability methods.

**Theoretical Claims:**

N/A

---

> ### Author Rebuttal · Authors · 2025-03-30
>
> Thanks for your comments! We address them point-by-point below. **In our responses to other reviewers, we provide additional human evaluations on our LLM judges and updated, much higher quality SAE concepts**.
>
> > **Q1**: Please define $n$ in section 4 notation.
>
> **A1**: Thank you for the suggestion! In Section 4, we use $n$ to denote the number of tokens in a given sequence. We will clarify this in our next revision.
>
> > **Q2**: How do you construct the instruction pool related to a given genre (lines 151 and 152)?
>
> **A2**: Due to space limitations, we provide details in Appendix I linked from section 3.1. For text-based instructions, we sample from Dolly-15K. For code-based instructions, we sample from a collection of Python-code puzzles formatted in an Alpaca-style (i.e., instructions paired with corresponding responses). For math-based instructions, we sample from GSM8K. With additional page allowance, we will ensure these details are included in the next revision.
>
> > **Q3**: The synthetic dataset constructed in Section 3.1 is only used for concept detection, not model steering, correct?
>
> **A3**: Yes, that is correct—it is used exclusively for concept detection. For model steering, we use existing instructions and concepts.
>
> > **Q4**: The procedure for concept detection from gradient-based methods described in Appendix H doesn't really make sense. I am not sure how using max-pooling on token importance for a given concept can be used for concept detection.
>
> **A4**: Thank you for raising this point! We agree that aggregating gradients at the sequence level might provide additional insights into gradient-based sensitivity methods. The main reason for applying max-pooling for IxG and IG is to maintain consistency with the setup used in other methods, ensuring a fair comparison. If resources permit, we will consider adding IxG and IG baselines that aggregate sequence-level gradients via sum or mean in the next revision.
>
> > **Q5**: For prompting, it's unclear why you needed different methods for concept detection and concept steering in the prompting case—it seems that both are somewhat the same task.
>
> **A5**: Thank you for the question. For concept detection, we use a template-based prompt asking the model, “Do you think the concept is incorporated in the given sentence?” (see Appendix J.3 for the full prompt). For model steering, we use gpt-4o-mini (acting as our prompt engineer) to generate a steering prompt (e.g., “You must include terms related to planting Apple trees in your responses. Here are some examples...”), which is quite different from our concept detection prompt.
>
> > **Q6**: In line 285, you mentioned, "we sample 10 instructions. We generate up to 128 tokens for each instruction over 14 steering factors." What does it mean to steer over 14 steering factors?
>
> **A6**: Thank you for raising the question. For model steering methods, we follow an existing paradigm in which we insert activation addition interventions into the model’s forward pass as follows:
>
> $ h_{Steer} = h_{Original} + \alpha \cdot w_{Steer} $
>
> Here, $\alpha$ represents the steering magnitude, and $w_{Steer}$ is the learned steering vector for a given method. As suggested by previous work, $\alpha$ is treated as a hyperparameter, and different values (steering factors) must be tested to select the optimal one for final evaluation. In our work, we use 14 different settings for $\alpha$ across all methods, and for each setting, we sample 10 instructions to steer. Due to space limitations, the discussion on steering factors is included in Appendix K. With additional page allowance, we will include further details in the main text in our next revision.

---

> > ### Comment · Reviewer_pwiS · 2025-04-03
> >
> > Thank you for the response and for the clarification. My recommendation remains as is; I think it's a very good benchmark and recommend acceptance.

---

### Official Review · Reviewer_ScQD · 2025-03-13

**Overall Recommendation:** 4

**Summary:**

This work develops a benchmark for testing different concept detection and model steering methods in LLM representation spaces. They test many different types of methods such as SAEs, finetuning, and prompting on their benchmark, including novel methods and novel applications of existing methods. As noted in the title, SAEs perform less well than many other methods, especially in model steering.

## Update after rebuttal

The authors have addressed points raised in my rebuttal, and included nice additional empirical results for human eval, zero-shot prompting, and different SAE labels. These are nice additions to an already great paper, and I recommend acceptance.

**Claims And Evidence:**

Yes, the submission uses solid empirical evidence to support its claims.

**Essential References Not Discussed:**

n/a

**Experimental Designs Or Analyses:**

I read the experimental details in the main paper, and some of the details in the supplementary (e.g. on hyperparameters). The experimental setup seems reasonable and good.

**Methods And Evaluation Criteria:**

The dataset creation procedure and steering / concept detection methods tested make a lot of sense. I would like to see more examples or other quality checks of generated data / judged responses. For instance, to what extent do the judgements agree with some trusted judgements (human or otherwise).

Many concept detection / steering methods are considered, and a novel method is even developed. This is impressive, especially given that slight modifications have to be made to some of them, and many different hyperparamters have to be searched.

**Other Comments Or Suggestions:**

n/a

**Other Strengths And Weaknesses:**

Strengths:
* Many other interesting related experiments in the appendix, such as learning to predict new concept subspaces.
* Ablations make the work more complete, e.g. comparing addition vs. clamping for SAE features, ablating $\alpha$ (it is interesting that high and low $\alpha$ have similar effects).

Weaknesses:
* Would be good to be clearer on the details of the prompting baseline in the main paper. This two-stage prompting technique is probably a bit more powerful than zero-shot prompting. It would be great to see how zero-shot prompting with a simple template works.
* Would prefer to see more examples as in Appendix N. The concept shown there is a bit vague, e.g. the first ReFT-r1 "failed to inject concept" response could arguably havea concept score of 2.

**Questions For Authors:**

Could you comment on concept discovery? And for instance the utility of SAEs or other methods for this? For instance, the concepts you consider in this paper are SAE features in the first place. Also, how would you expect the fact that concepts are taken from Gemma SAE features to affect your empirical results?

**Relation To Broader Scientific Literature:**

This work is highly relevant to mechanistic interpretability and model steering literatures. It calls into question the utility of certain mechanistic interpretability techniques in model steering and even concept detection.

**Theoretical Claims:**

n/a. this is mostly an empirical work.

---

> ### Author Rebuttal · Authors · 2025-03-30
>
> Thanks for your comments! We address them point-by-point below. **We supply preliminary human evaluations as well as results for a new prompting baseline.**
>
> > **Q1: Performing a proper human evaluation to ground AxBench.**
>
> **A1**: Great suggestion! We conducted a preliminary human evaluation by recruiting 5 participants, each of whom rated 30 steering generations from different methods (e.g., Prompt, ReFT‑r1, SAE) using the same scoring system as our LLM judges. We then computed two types of Pearson correlation coefficients to assess agreement:
> - **Human-Human Agreement:** For each pair of human participants, we computed the Pearson correlation between their ratings, applied Fisher’s Z transformation, averaged them in the transformed space, and converted the result back to a Pearson correlation coefficient, yielding a score of **0.57**.
> - **LLM-Human Agreement:** We also computed the Pearson correlation between the LLM’s ratings and each human’s ratings, applied Fisher’s Z transformation, and averaged these values across the 5 participants, yielding a score of **0.58**.
> These measures indicate that the LLM behaves much like just another human participant in terms of rating consistency. Given that steering is a highly subjective task about which individuals frequently disagree, these strike us as high correlation numbers overall. Moreover, prior work on similar annotation tasks [[1]](https://arxiv.org/abs/2406.06369)[[2]](https://arxiv.org/abs/2204.00447) has reported correlations of at most 0.6, further validating our findings. We have uploaded our anonymized human annotation interface [here](https://tinyurl.com/2xyd44d).
>
> > **Q2**: Would be good to be clearer on the details of the prompting baseline in the main paper. It would be great to see how zero-shot prompting with a simple template works.
>
> **A2**: This is a great suggestion! With additional page allowance, we will further clarify our prompting baseline in the main text. We also ran additional experiments with zero-shot prompting, using a template-based approach that directly asks the model to incorporate a concept in its response. Below are the results:
>
> | Method               | 2B L10 | 2B L20 | 9B L20 | 9B L31 | Avg.  |
> |----------------------|-------------------|-------------------|-------------------|-------------------|-------|
> | Prompt               | 0.731             | 0.744             | 1.081*            | 1.062*            | 0.905*|
> | Prompt (Zero-shot)   | 0.735             | 0.727             | 1.051             | 1.016             | 0.882 |
>
> where * means statistically significant. As expected, zero-shot prompting performs worse than LLM-aided prompting. The gap between these two variants is surprisingly small, indicating that instruction-tuned models can be easily steered without sophisticated prompt engineering.
>
> > **Q3**: Would prefer to see more examples as in Appendix N.
>
> **A3**: We will provide additional examples and improve their quality in our next revision.
>
> > **Q4**: Could you comment on concept discovery?
>
> **A4**: Although SAEs lag far behind supervised methods in empirical results, we believe they serve as excellent “hypothesis generators” for concept discovery in an unsupervised manner. However, these hypotheses must be causally verified through intervention-driven experiments, as we did for model steering in AxBench. The concepts identified by SAEs can naturally become learning targets for supervised methods, which yield better subspaces. On the other hand, supervised methods can utilize any concept list—not just those derived from SAEs.
>
> > **Q5**: How would Gemma SAE features to affect your empirical results?
>
> **A5**: Great point! We agree that the labels from the auto-interpretability pipeline contain noise. To validate our pipeline, we ran preliminary experiments using a set of high-quality SAE labels released recently [[1]](https://arxiv.org/abs/2501.08319) for both SAE and ReFT‐r1, and we observed similar performance patterns with the Gemma-2-2B model (i.e., ReFT-r1 is significantly better than SAE). Interestingly, SAE steering performance increased slightly, suggesting that better feature annotations yield improved steering results.
> Below are the results:
>
> | Method             | 2B L10 | 2B L20 |
> |--------------------|-------------------|-------------------|
> | SAE (original)     | 0.175             | 0.175             |
> | SAE (new)          | 0.25              | 0.25              |
> | ReFT‐r1 (original) | 0.55              | 0.50              |
> | ReFT‐r1 (new)      | 0.59              | 0.45              |
>
> Additionally, SAE’s performance on concept detection increased from 0.75 to 0.85 for both layers. In summary, **higher quality concept labels improve SAE’s performance without closing the gap to other supervised methods**. We will incorporate these newer findings into our next revision. Note that the original and new results are based on around 20 concepts.

---

> > ### Comment · Reviewer_ScQD · 2025-04-02
> >
> > We thank the authors for their thorough rebuttal. These additional results (human eval, zero-shot prompting, and different SAE labels) add to the paper. I think this is a great paper, and recommend acceptance.

---

### Official Review · Reviewer_eN7w · 2025-03-14

**Overall Recommendation:** 5

**Summary:**

This paper proposes a new large-scale benchmark for steering and interpretability, with reported results on Gemma2. They find that prompting outperforms all existing interpretability and steering methods, followed by fine-tuning, for steering. They also find that SAEs are not competitive for either task. The authors propose a novel adaptation of Representation Finetuning called ReFT-r1 for steering models in an interpretable manner, with results validating its utility. Finally, the authors publicly release trained supervised dictionaries along with their benchmark.

**Claims And Evidence:**

The claims made in the paper are supported by very clear and convincing evidence.

**Essential References Not Discussed:**

N/A

**Experimental Designs Or Analyses:**

The soundness and validity of the experimental designs and empirical results were all validated. No issues were found beyond what was discussed in "Methods And Evaluation Criteria".

**Methods And Evaluation Criteria:**

The proposed benchmark dataset along with the evaluation metrics are highly intuitive for the task at hand.
- One concern is that the features chosen are the labels the authors found for the evaluated SAEs on Neuronpedia. However, these SAEs were labeled with an (admittedly barely) outdated auto-interpretability method. Thus, many of the labels may have been erroneous or, not specific, or generally irrelevant; however, they were still the features by which all methods were compared to. It would be beneficial for the authors to further discuss the limitations of these features, and whether other concepts may be important or interesting to consider and evaluate for as well.
- Furthermore, the authors may want to include discussion of how the choice of layer may affect the results. Perhaps intervention in later layers of the model would have been more effective after the model has transitioned to the next token prediction task and can be steered towards predicting tokens related to the desired behavior.
- Finally, the authors may want to elaborate on how feature sets should be generated for training supervised dictionaries without having to first train an SAE, label it, and then use those labels as the feature set. Is there some unsupervised method to generate relevant features in an automatic manner?

**Other Comments Or Suggestions:**

N/A

**Other Strengths And Weaknesses:**

Strengths:
- This paper is well-written, easy-to-understand, and thorough in its evaluation.
- The results are novel and highly relevant to the community, with implications on future research on model steering and sparse autoencoders.

Weaknesses:
- This is not a significant weakness, but given the breadth of methods evaluated, I am curious why the authors stuck primarily to linear probes. It would have been interesting to see a comparison with nonlinear probes, where steering can be performed by increasing the activation while minimizing distance from the original representation.
- The authors did not validate their LLM-as-a-judge setup. Given that this setup seemed to be highly unstable in other configurations, a small-scale validation would be beneficial.

**Questions For Authors:**

N/A

**Relation To Broader Scientific Literature:**

The key contributions of this paper relate to broader literature in mechanistic interpretability and model steering, particularly with respect to literature on sparse autoencoders, as this paper empirically shows that SAEs are limited in their capacity to steer and control models, as well as to predict the presence of concepts in latent representations. This paper is beneficial to the community in that it sets a more rigorous standard for thoroughly evaluating methods on desired downstream tasks and benchmark against existing methods.

**Theoretical Claims:**

This is not applicable as no theoretical claims or proofs were given in the paper.

---

> ### Author Rebuttal · Authors · 2025-03-30
>
> Thanks for your comments! We address them point-by-point below. **We provide preliminary results by using a set of high-quality SAE labels released recently below.**
>
> > **Q1**: One concern is that the features chosen are the labels the authors found for the evaluated SAEs on Neuronpedia. However, these SAEs were labeled with an (admittedly barely) outdated auto-interpretability method.
>
> **A1**: Great point! We agree that the labels from the auto-interpretability pipeline contain noise. **To validate our pipeline, we ran preliminary experiments using a set of high-quality SAE labels released recently** [[1]](https://arxiv.org/abs/2501.08319) for both SAE and ReFT‐r1, and we observed similar performance patterns with the Gemma-2-2B model (i.e., ReFT-r1 is significantly better than SAE). Interestingly, SAE steering performance increased slightly, suggesting that better feature annotations yield improved steering results.
> Below are the results:
>
> | Method             | 2B L10 | 2B L20 |
> |--------------------|-------------------|-------------------|
> | SAE (original)     | 0.175             | 0.175             |
> | SAE (new)          | 0.25              | 0.25              |
> | ReFT‐r1 (original) | 0.55              | 0.50              |
> | ReFT‐r1 (new)      | 0.59              | 0.45              |
>
> Additionally, SAE’s performance on concept detection increased from 0.75 to 0.85 for both layers. In summary, **higher quality concept labels improve SAE’s performance without closing the gap to other supervised methods**. We will incorporate these newer findings into our next revision. Note that the original and new results are based on around 20 concepts.
>
> > **Q2**: Furthermore, the authors may want to include discussion of how the choice of layer may affect the results.
>
> **A2**: Yes, the choice of layer does seem to affect steering performance. For example, in larger 9B models, steering at earlier layers (e.g., L20 vs. L31) appears to be advantageous, which is not as evident for smaller 2B models. While later layers might be more effective for generating tokens, they involve fewer downstream layers, potentially reducing the expressivity of the steering vector.
>
> > **Q3**: Finally, the authors may want to elaborate on how feature sets should be generated for training supervised dictionaries without having to first train an SAE. Is there some unsupervised method to generate relevant features in an automatic manner?
>
> **A3**: Great suggestion! Yes, this would be an interesting extension study. For supervised dictionary learning methods (SDLs), obtaining labels from SAEs is not required; we use SAE labels solely to maximize SAE performance. One promising direction is to compile expert-level concept lists (e.g., reasoning-related concepts such as "backtracking") and then use SDLs to identify effective steering vectors for both concept detection and model steering.

---

### Official Review · Reviewer_xuXA · 2025-03-22

**Overall Recommendation:** 3

**Summary:**

- Authors introduce AxBench, a benchmark to evaluate LLM steering, ie. capability at following instruction. In that context, authors evaluate multiple representation-based techniques, including sparse auto-encoders and linear probes, to achieve that goal. The findings show that representation techniques are still far behind simple prompting and traditional fine-tuning. In parallel, authors introduce a new steering technique, called ReFT-r1, that shows improvement upon other representation-based techniques, but with still a gap compared to prompting and fine-tuning.

**Claims And Evidence:**

- There are multiple claims made in this work. A. "There is no benchmark for making direct comparisons between" a variety of representation-based techniques. B. Representation steering, and sparse autoencoders among other, "is still far behind simple prompting and finetuning baselines".
- While claim B. is supported by multiple experiments through the paper, with experiments done on Gemma2 2B and 9B, the claim A is problematically not supported in the paper. The Section 2 related work does not make any mention of concurrent benchmarks to evaluate model instruction following, and there is a mix of the two contributions (AxBench and ReFT-r1) through the overall paper.
- For instance, Figure 2 is clealry mixing the two topics: the left and middle part are about AxBench, while the right is about steering techniques.

**Essential References Not Discussed:**

- The overall work focuses only on the single benchmark AxBench as introduced by the author. One could have appreciated if the authors could link their work with other instruction following benchmarks. There are many, including the famous IFEval paper that is surprisingly not mentioned in this work.

**Experimental Designs Or Analyses:**

- One could have appreciated reporting results on other benchmarks, beyond just AxBench. Or alternatively, performing a proper human evaluation to ground AxBench. Otherwise, this makes difficult to verify the claims in this work. AxBench is synthetically built and evaluated, and no point of comparison with other benchmarks are provided.

**Methods And Evaluation Criteria:**

- Authors use Gemma-2 as base for their experimental settings. The AxBench benchmark evaluates both Concept Detection and Model Steering. Authors execute a quick deep dive into the experimental details

**Other Comments Or Suggestions:**

- The overall paper is not easy to read, with many abbrevations such as SDL and SAE. In particular, SAE is only introduced in L058 where it merely mentions it to be about unsupervised methods, but it is only latter in the document that one can assume that SAE are sparse autoencoders. SDL in L058 too hasn't any references attached to it. And the rest of the section of 2 "Related Work" does not give more details about SDL.
- More problematic perhaps than abbrevations, the paper seems to not be able to pick which contribution to highlight between AxBench and ReFT-r1. For instance, the title of the paper is about 'AxBench' but the related work section only focuses on steering techniques, lacking any references at other instruction following benchmarks and why they might fall short. Findings as well reported in the conclusion continue mixing both concepts: on one had ReFT-r1 is not as good as prompting, but on the other hand "No matter the outcome, (...) evaluation benchmarks like AxBench are necessary (...).". It is not clear what is the value proposition here. A clear articulation with the SOTA seem to be needed here. Either this paper is about ReFT-r1, and then one could hope having competitive results compared to other methods, or it is about AxBench (as pointed by the title), and then one could hope having a clear articulation with other Instruction Following Benchmarks.
- The paper deeps dive quickly into AxBench, and fails to connect it to other benchmarks or human validation. One could have appreciated a better articulation with the rest of the literature.
- The legend of Figure 2 is bit confusing. It mentions "(a)", "(b)" and "(c)", but those elements are not reported in the actual figure. Besides, it mixes AxBench (the actual benchmark) with explanations about how two familiies of steering techniques work.

**Other Strengths And Weaknesses:**

- The work proposes a deep dive analysis on the introduced AxBench, and how it helps measuring model 'steering-ness'. However, one weakness of this work is to fail to connect this in-depth analysis with other benchmarks, such as the widely adopted IFEval, and with human evaluation, i.e AxBench evaluation is synthetically created (Concept Detection) and synthetically evaluated (Model Steering).

**Questions For Authors:**

Dear authors, thank you for your work:
- Could you please describe what is the main significant contribution of this work? It seems two contributions are mixed: AxBench (from the title) and ReFT-r1 (from the abstract). However, I found difficult to follow the articulation with the rest of the literature. The Section 2 "Related Work" is only about other steering techniques, and ignores all the literature about Instruction Following Benchmarks. On the other hand, the title of the work starts with the token "AxBench", which makes the reader expecting a Benchmark paper (like IFEval, MIABench, M-IFEval, etc.). Could you please clearly position this work? Is that a work that contributes a new benchmark, and hence Section 2 should be probably re-written, or is that a work about ReFT-r1, in such case the title is probably not adequate.

**Relation To Broader Scientific Literature:**

- Surprisingly given the title of this paper, the Related Work section is only about steering techniques, and no other instruction following benchmarks are discussed or mentioned. This is problematic and makes hard to understand this paper contributions. See other sections of this review.

**Theoretical Claims:**

See 'Claims and Evidence'

---

> ### Author Rebuttal · Authors · 2025-03-30
>
> Thanks for your comments! We address them below. **We articulate our goal and how AxBench differs from IFEval-like benchmarks, and supply preliminary human evaluations.**
>
> > **Q1**: The Section 2 related work does not mention any concurrent benchmarks to evaluate model instruction following (e.g., IFEval). The paper dives quickly into AxBench and fails to connect it to other benchmarks.
>
> **A1:** We want to clarify the purpose of creating AxBench. First, **AxBench is not an instruction following benchmark for LLMs.** IFEval is intended to compare the instruction-following abilities of LLMs using *a small set of easily verifiable metrics*. AxBench is *an open-vocabulary evaluation benchmark* for concept-based steering and detection, which compares the efficacy of different steering/detection methods on a single LLM, and is not verifiable with simple rules.  **Our concepts can be highly abstract** and are described using natural language descriptions (i.e., this is not comparable to asking LLMs to respond with X tokens or include keywords Y, as in IFEval). **Unlike IFEval, a rule-based evaluation metric is impossible for such open-ended concepts.** Moreover, **our concepts are not artificially created, and are grounded with interpretability methods**. They emerge in LLM's representations and are found in an unsupervised manner with SAE [[1]](https://arxiv.org/abs/2408.05147) [[2]](https://arxiv.org/abs/2410.13928). Another benefit of doing so is having access to thousands of such concepts, so that we can evaluate steering methods at scale. We will clarify the distinction between steering and instruction following in our next revision.
>
> > **Q2**: Performing a proper human evaluation to ground AxBench.
>
> **A2**: Great suggestion! We conducted a preliminary human evaluation by recruiting 5 participants, each of whom rated 30 steering generations from different methods (e.g., Prompt, ReFT‑r1, SAE) using the same scoring system as our LLM judges. We then computed two types of Pearson correlation coefficients to assess agreement:
> - **Human-Human Agreement:** For each pair of human participants, we computed the Pearson correlation between their ratings, applied Fisher’s Z transformation, averaged them in the transformed space, and converted the result back to a Pearson correlation coefficient, yielding a score of **0.57**.
> - **LLM-Human Agreement:** We also computed the Pearson correlation between the LLM’s ratings and each human’s ratings, applied Fisher’s Z transformation, and averaged these values across the 5 participants, yielding a score of **0.58**.
> These measures indicate that the LLM behaves much like just another human participant in terms of rating consistency. Given that steering is a highly subjective task about which individuals frequently disagree, these strike us as high correlation numbers overall. Moreover, prior work on similar annotation tasks [[3]](https://arxiv.org/abs/2406.06369)[[4]](https://arxiv.org/abs/2204.00447) has reported correlations of at most 0.6, further validating our findings. We have uploaded our anonymized human annotation interface [here](https://tinyurl.com/2xyd44d).
>
> > **Q3**: Some Appendix could be probably removed. The overall paper is not easy to read.  The legend of Figure 2 is bit confusing.
>
> **A3**: We'll clarify our notations early and update Figure 2 and the Appendix in our next revision.
>
> > **Q4**: The paper seems to not be able to pick which contribution to highlight between AxBench and ReFT‐r1.
>
> **A4**: **Our goal is to evaluate whether existing representation-based steering methods are competitive in terms of concept detection and model steering compared to finetuning or prompting methods.** Most of our approaches produce rank-1 steering vectors, which is the simplest steering setting designed to minimize inference-time overhead while rivaling prompting in practical scenarios.
>
> To accomplish this goal, our contributions are: (1) **AxBench**, which, to the best of our knowledge, is the first benchmark for evaluating representation-based steering methods at scale; and (2) **ReFT‐r1**, a representation-based method derived from ReFT [[5]](https://arxiv.org/abs/2404.03592) that rivals finetuning or prompting methods. As shown in our Figure 1, without ReFT‐r1, representation-based methods significantly lag behind, making it nearly hopeless for them to catch up. **We believe that our two contributions work organically together** to provide a solid foundation for evaluating methods and paving the way forward for representation-based approaches.
>
> Our work shows that while SAEs were once seen as the only scalable method for training steering vectors, ReFT‐r1 disproves this for concept detection and model steering. AxBench is essential to evaluate interpretability methods as they scale against finetuning and prompting. Although ReFT‐r1 still lags behind, we think it provides important insights into how to train better steering vectors that can be useful in practice.

---

### Decision · Program_Chairs · 2025-05-01

**Decision:**

Accept (spotlight poster)

**Comment:**

This work presents a new benchmark for steering methods coined AxBench. They evaluate 2 axes: **concept detection** (using a held-out dataset per concept and measuring AUROC) and **model steering** (where long-form generations are evaluated with an LLM-as-a-judge). Their findings show that steering methods are still behind prompting specially in model steering, and that fine-tuning does not reach prompting performance but can improve over steering. The authors additionally propose a new steering method based on ReFT, coined ReFT-r1 that narrows the gap with prompting. They also explore SAEs, showing that they do not perform as well on any of the axes checked in AxBench.


All the reviewers agree in the value of this work for the community.


I believe this work should be presented at ICML, given the current trend on exploring steering methods. There is a lack of benchmarks in the literature, and AxBench could enable a more comprehensive evaluation of past and new steering proposals.